# Executable Functional Abstractions: Inferring Generative Programs for Advanced Math Problems

## Abstract

Abstract Interpretation provides a framework for approximating the behavior of discrete systems by establishing a correspondence between concrete execution traces and abstract properties. We apply this framework to mathematics to address the inverse problem: automatically synthesizing a general program (the *abstraction*) from a single concrete example, which executes to produce specific, valid problem instances (the *concretization*). Prior approaches to capturing this structure rely on hand-crafted templates, a labor-intensive process that restricts the technique to arithmetic word problems or small datasets. We introduce `EFAGen`, a method that operationalizes this inference as a program synthesis task, generating Executable Functional Abstractions (EFAs) that encode the parameters, constraints, and solution procedure of the seed problem. Because formal verification of synthesized code is intractable, we filter candidates using executable unit tests that enforce necessary properties. We demonstrate that these inferred abstractions enable data augmentation that complements existing strong data mixes for math reasoning and facilitate adversarial search to discover problem variants that models fail to solve.

## 1 Introduction

Abstract Interpretation (Cousot & Cousot, 1977) provides a rigorous framework for approximating the behavior of discrete systems. It establishes a correspondence between a *concrete domain* of specific execution traces and an *abstract domain* of general properties. In the context of mathematics, we can view a specific problem instance (e.g., "Find the GCD of 6 and 126") as a point in the concrete domain. The underlying general logic (variables, constraints, and solution procedure) serves as its representation in the abstract domain. We term this programmatic representation an **E**xecutable **F**unctional **A**bstraction (`EFA`). Possessing the `EFA` for a problem is powerful. It allows one to analyze the general class of the problem rather than a single instance and enables the generation of infinite valid variants through *concretization* functions. These variants have the potential to be useful as a source of training data or to construct challenging benchmarks for evaluation.

However, a reliable "abstraction function", a mechanism to automatically lift a concrete problem into a valid `EFA`, does not exist for complex mathematics. Current approaches to obtaining these abstractions, such as GSM-Symbolic (Mirzadeh et al., 2025) and FnEval (Srivastava et al., 2024), rely heavily on manual engineering. Humans must painstakingly identify variables, define domains, and write code for every problem template. This manual reliance restricts the abstractions to simple grade-school arithmetic or small, curated datasets is not scalable. Constructing an abstraction function for complex mathematics poses two fundamental challenges. First, synthesis is difficult: identifying the correct parameters, discovering non-trivial constraints, and generalizing the solution logic must all succeed simultaneously. Getting any component wrong yields an invalid abstraction. Second, verification is intractable: formally proving correctness of these synthesized programs is beyond current capabilities.

Our key insight is to reformulate this open-ended inference problem as a tractable search problem defined by executable code and operational verification. We introduce `EFAGen`, which operationalizes the Abstract Interpretation relationship as a program synthesis task. Our inference pipeline acts as an *abstraction function* $\alpha$. It takes a concrete problem instance $x$ as input and synthesizes an `EFA` that

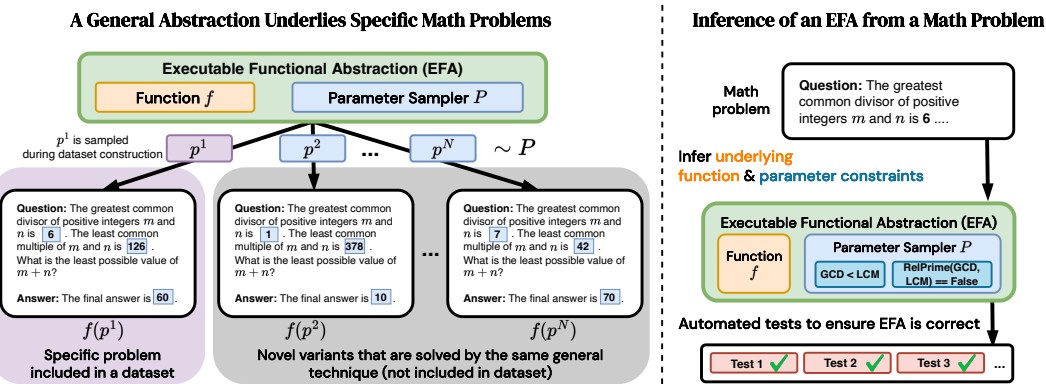

Figure 1: **Left**: We view math problems through the lens of Abstract Interpretation: specific problem instances with concrete values lie in the *concrete domain*, while **executable functional abstractions (EFAs)** represent the *abstract domain* of parameterized logic and constraints. The *concretization function* $\gamma$ (via `sample()`) generates valid concrete instances from an EFA. **Right**: We study the task of automating the *abstraction function* $\alpha$ that lifts a concrete problem instance into its corresponding EFA, automatically inferring parameters, constraints, and general solution procedures from natural language problems. We approach this as a program synthesis task, and show the validity of the inferred EFAs as well as their utility in downstream tasks.

transforms specific numerical values into typed parameters, encodes constraints between them, and implements a general solution procedure valid for any parameterization satisfying these constraints. Each synthesized EFA implements a *concretization function* $\gamma$ via its `sample()` method, which instantiates the abstract schema into concrete problem instances. Rather than formal verification, we implement *operational soundness checks* as executable unit tests that verify necessary conditions for validity. These checks ensure that $x \in \gamma(\alpha(x))$—the abstraction can reproduce the original instance— but also that sampled variants are non-trivial (distinct from the seed), solvable (match expected answers), and valid (satisfy domain constraints). We generate multiple candidate programs using an overgenerate-and-filter approach (Li et al., 2022), treating each EFA as a hypothesis and selecting those that pass all operational checks. This search procedure enables us to discover abstractions that are operationally sound with respect to the seed problem.

We confirm the internal validity of the inferred EFAs by measuring the faithfulness of the generated variants to the seed problem and their utility in training models. We then demonstrate the applications of EFAs to two downstream tasks. Specifically, we show that EFAs can be used for adversarial search to discover harder problem variants and for data augmentation. In the latter, we demonstrate that EFA-generated data is high quality and complementary to existing data augmentation methods. Our experiments show that EFA-based augmentation combined with NuminaMath (**?**) yields better performance than using NuminaMath alone. This suggests that the inferred abstractions capture structural patterns distinct from those in standard corpora.

We make the following contributions.

- We formalize EFAs and develop EFAGen, an automated approach to infer executable abstractions from competition-level math problems by treating abstraction as a synthesis and verification task.

- We demonstrate that the execution feedback from our validity tests acts as a reward signal. This enables LLMs to self-improve at the abstraction task via reinforcement learning.

- We empirically show that inferred EFAs provide a complementary data source to strong baselines like NuminaMath. Data augmentation with EFAs improves performance on MATH-500 and FnEval, and EFAs can be used to search for easier harder or easier variants of problems.

## 2 EXECUTABLE FUNCTIONAL ABSTRACTIONS (EFAS)

Our goal is to automatically convert math problems with static numerical values into **parameterized abstractions** that can generate variants of the original problems. We refer to these parameterized abstractions as **Executable Functional Abstractions (EFAs)**. EFAs enable the systematic generation

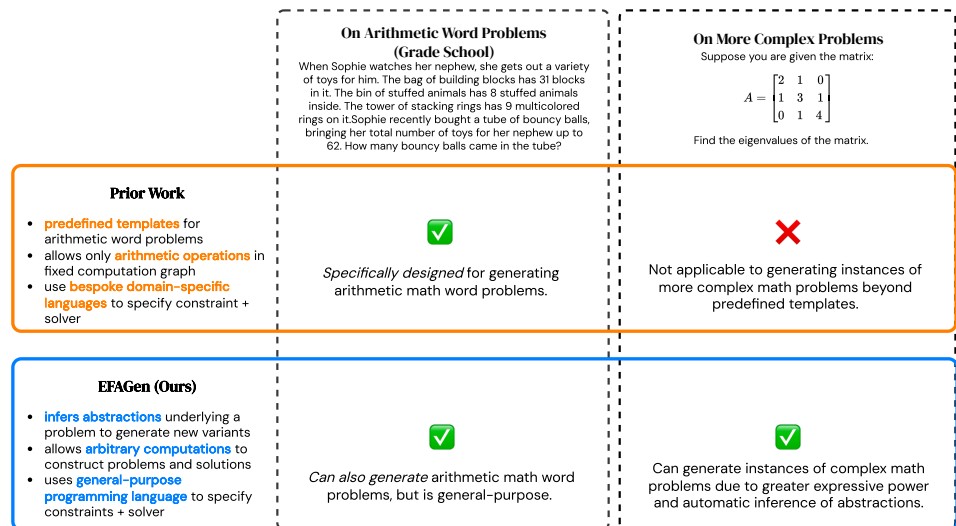

Figure 2: **EFAGen generalizes prior work on constructing arithmetic word problems to automatically constructing more complex, higher-level math problems**. Given a math problem and solution, EFAGen infers an underlying abstraction whose construction and general solution may involve arbitrary computations beyond fixed sequences of arithmetic operations. For example, the abstraction underlying the eigenvalue problem on the right is that of a tridiagonal $3 \times 3$ matrix. The general solution requires a symbolic computation composed with a numerial root-finding procedure. Details of inferred EFA code in Fig. 7.

of new problem instances by varying numerical parameters while preserving the underlying problem-solving logic. We operationalize the task of inferring an EFA for a static math problem as a program synthesis task where the goal is to write a class implementing the EFA. We use LLMs to generate many candidate EFA implementations for a static problem and use a suite of automatic unit tests to filter the candidates by rejecting mathematically unsound ones. Below, we describe the desired properties of EFAs (Sec. 2.1), how an EFA is represented as a Python class (Sec. 2.2), and how we infer EFAs from static math problems using LLMs (Sec. 2.3).

## 2.1 DESIRED PROPERTIES OF ABSTRACTIONS

An effective abstraction of a math problem must support variation, preserve validity, and enable automated problem-solving. We identify three core properties of an EFA:

- **Structured parameter space**: The abstraction should define a set of parameters that characterize the problem and specify valid relationships among them. This includes identifying which parameters are independent, how dependent parameters are derived, and what constraints must be satisfied to ensure valid problem instances. Such structure enables systematic variation, ensuring that changes to parameters yield meaningful variants with potentially different solutions.
- **Procedural generation of instances**: The abstraction should support random sampling of a set of valid parameters (e.g., EFA.sample() in Sec. 2.2) and converting the abstract problem into natural language form (e.g., EFA.render() in Sec. 2.2), to help users generate valid problem instances by sampling parameter values within defined constraints. These constraints are problem-specific and crucial for generating diverse but coherent examples.
- **Executable solution logic**: The abstraction should include a method (e.g., EFA.solve() in Sec. 2.2) that computes the correct answer for any valid parameter configuration. This solution logic is typically derived from the chain-of-thought (Wei et al., 2022) used for the static version of the problem and can be implemented as an executable program.

## 2.2 EFA AS A PYTHON CLASS

As shown in Fig. 3(a), each EFA is implemented as a Python class that contains the logic of a math problem in a parameterized form. The class defines a list of parameters along with three key methods:

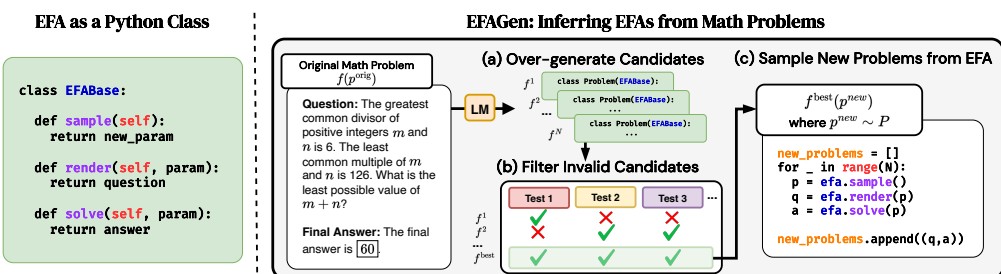

Figure 3: **Left: Representation of an executable functional abstraction (`EFA`)** as a Python class. **Right: Overview of `EFAGen`, a method for automatically inferring `EFAs`** from a math problem. In `EFAGen`, we (a) over-generate multiple `EFA` candidates with an LLM and (b) filter out invalid candidates that fail automated tests. The `EFA` can generate new problem variants by sampling parameters and executing the solver. Full code is in Appendix E.

- **`EFA.sample() → parameters`**: Samples a valid set of parameters representing problem variants, respecting all constraints specified in the abstraction.
- **`EFA.render(parameters) → question`**: Provides a natural language problem statement, given a specific (sampled) parameter set. This ensures that each generated instance is presented in a format suitable for human or model consumption. In most cases, this involves reusing the problem statement of the seed instruction and mutating the numerical values to be consistent with the given parameters.
- **`EFA.solve(parameters) → answer`**: Computes the correct answer for a given parameter configuration. The solution is expressed as a numerical expression derived through deterministic computations over the parameters. The solver does not need to access the natural language problem statement, as the solution is only dependent on the parameterization of the problem, which is a structured object.

These methods operationalize the abstraction and enable automated generation, rendering, and evaluation of math problems.

## 2.3 `EFAGEN`: INFERRING `EFAS` FROM MATH PROBLEMS

We introduce `EFAGen`, a framework for automatically constructing `EFAs` from static math problems. Given a problem statement and its solution procedure (typically expressed as chain-of-thought reasoning), `EFAGen` uses a large language model (LLM) to generate a candidate `EFA` implementation that captures the logic and structure of the original problem. This process relies on supervision that is readily available in many math datasets.

Since generating correct and robust code is challenging for LLMs, `EFAGen` adopts an overgenerate-and-filter approach inspired by AlphaCode (Li et al., 2022). As described in Fig. 3 (a), for each problem, we sample $N$ (e.g., 50) `EFA` candidates and apply a suite of automated tests to discard invalid abstractions. `EFAGen` uses the following tests to validate candidate `EFAs`, as illustrated in Fig. 3 (b):

- **`is_extractable(response)`**: Verifies that the class contains all required methods.
- **`is_executable(EFA)`**: Confirms that the class can be instantiated and executed without errors, and methods like `EFA.sample()` and `EFA.solve()` can be called without errors.
- **`has_dof(EFA)`**: Ensures that sampled parameters differ, rejecting `EFAs` with zero degrees of freedom that cannot produce new problems.
- **`is_single_valued(EFA)`**: Confirms that identical parameters yield equivalent solutions, rejecting impermissible implementations including multivalued functions or incoherent abstractions.
- **`matches_original(EFA, orig_params, orig_sol)`**: Validates that the abstraction, when instantiated with the original parameters, produces the original problem and solution. This serves as a cycle-consistency or soundness check.

Any program that fails these tests cannot logically be a valid implementation of an `EFA`. `EFAGen` enables generation of `EFAs` at scale, as shown in Fig. 3 (c), as large numbers of candidate `EFAs` can

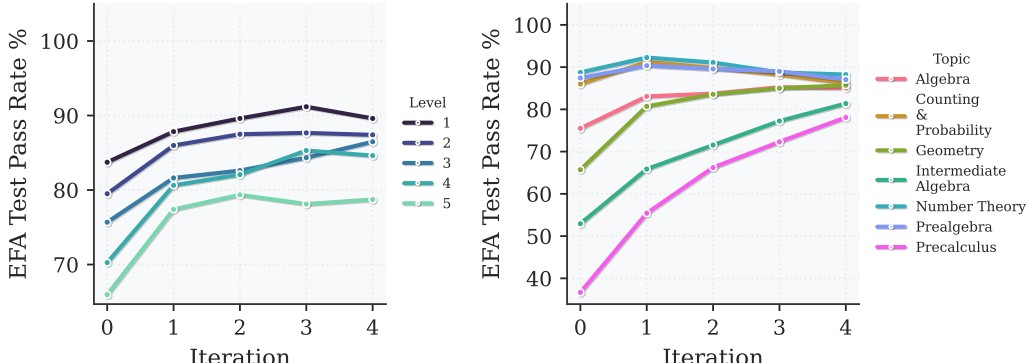

Figure 4: **LLMs can use our tests to self-improve at inferring `EFAs`.** We plot the percentage of constructed `EFAs` passing all tests across iterations of self-training, grouped by MATH problem difficulty (left) and by problem category (right). Harder difficulty levels and problem categories are harder to infer `EFAs` for and improve more during training.

be generated and filtered automatically. Over time, these tests can also be used to fine-tune LLMs toward better abstraction generation, such as with reinforced self-training (Singh et al., 2023; Dong et al., 2023) or reinforcement learning with verifiable rewards (Lambert et al., 2024).

## 3 EXPERIMENTS & RESULTS

Below, we show experiments on self-improving at inferring `EFAs` (Sec. 3.1), faithfulness (Sec. 3.2) and learnability (Sec. 3.3) of `EFAs`, the complementarity of `EFAs` with existing data generation methods (Sec. 3.4, Sec. 3.5). In the Appendix, we analyze the quality of `EFA`-generated data (Appendix B), scaling experiments (Appendix C), applying `EFAs` to find hard variants of problems (Sec. 3.7), ablations (Appendix D), and `EFAGen` inference on olympiad-level problems (Sec. 3.6).

**Datasets.** Throughout this section, we use the following datasets in our experiments:

- **MATH** (Hendrycks et al., 2021). Competition math dataset with a test set of 5k math problems described in text comprising different categories and five levels of difficulty. We show in Sec. 3.1 that LLMs struggle with task of `EFA` generation and we improve their performance by training on the `EFA` generation task using the MATH train set consisting of 7.5k problems.
- **FnEval** (Srivastava et al., 2024). A functional version of the MATH benchmark designed to evaluate generalization. It consists of multiple "snapshots", each containing variations of problems from the MATH dataset. These variations preserve the abstract reasoning structure of the original problems. We use two snapshots to test if our method can capture the underlying abstractions of a problem and generalize to unseen, related instances.
- **MATH-Hard** is a subset of MATH test problems of the highest difficulty (level 5) across all categories (1387 problems).

**Metrics.** To evaluate the performance of LLMs we use the following metrics:

- **EFA Success Rate.** We measure the ability of LLMs to generate valid, high-quality EFAs (defined in Sec. 2.1) as the frequency (%) of EFAs generated that past all the diagnostic tests (c.f. Sec. 2.3).
- **Pass@$k$ Rate (%).** Following Chen et al. (2021), we measure the ability of LLMs to solve math problems by sampling 25 generations with temperature sampling and estimating the unbiased pass@$k$ rate, i.e., the likelihood that out of $k$ generated solutions any one yields the correct answer.

### 3.1 SELF-IMPROVEMENT: LMS IMPROVE AT EFA INFERENCE WITH EXECUTION FEEDBACK

Inferring valid `EFAs` across diverse math problems is challenging, especially as the difficulty and complexity of topics increases. For instance, as shown in Fig. 4, Llama3.1-8B-Instruct (Llama Team, 2024) struggles to generate valid `EFAs` for Level 5 problems and for topics such as Precalculus in the

Table 1: **EFAs faithfully capture the solutions of the problems they were derived from (left), and problem variants constructed by `EFA`s share learnable structure (right).** Left: Giving solutions to problems variants from an `EFA` as in-context examples nearly doubles the solve rate of an LLM on the seed problem the `EFA` was derived from. Right: Giving solutions to problem variants from an `EFA` as in-context examples helps an LLM solve a holdout set of variants from the same `EFA`. See Sec. 3.2 and Sec. 3.3 for details.

| Faithfulness (Sec. 3.2): `EFA` helps on the original problem | | | Learnability (Sec. 3.3): `EFA` helps on its variants | | |
|---|---|---|---|---|---|
| Initial Pass@1 | +Data from `EFA` | Sample Size | Initial Pass@1 | +Data from `EFA` | Sample Size |
| 15.66 | 38.73 (+23.07%) | 307 | 14.58 | 31.23 (+16.65%) | 1,000 |

Table 2: **`EFA`s are effective at data augmentation.** Comparison with and without synthetic data augmentation using problems drawn from generated `EFA`s. The table shows performance across MATH-500 and FnEval benchmarks (November and December snapshots). When augmenting, we use a 1:1 ratio of examples drawn from training data vs. from an `EFA`, and report results using 33% of the MATH train set and 100% of the train set.

| Training Data | MATH-500 | | | FnEval (November Split) | | | FnEval (December Split) | | |
|---|---|---|---|---|---|---|---|---|---|
| | Pass @ 1 | Pass @ 10 | Maj @ 25 | Pass @ 1 | Pass @ 10 | Maj @ 25 | Pass @ 1 | Pass @ 10 | Maj @ 25 |
| MATH (33%) | 22.4 | 56.4 | 36.8 | 24.5 | 55.3 | 39.6 | 24.4 | 55.4 | 39.3 |
| +EFA (1:1) | 24.3 | 58.3 | 38.8 | 26.7 | 59.2 | 41.8 | 26.6 | 57.3 | 41.2 |
| | (+1.9%) | (+1.9%) | (+2.0%) | (+2.2%) | (+3.9%) | (+2.2%) | (+2.2%) | (+1.9%) | (+1.9%) |
| MATH (100%) | 24.3 | 57.8 | 37.0 | 26.8 | 58.6 | 43.1 | 26.5 | 57.6 | 41.5 |
| +EFA (1:1) | 26.1 | 60.6 | 40.4 | 29.3 | 60.1 | 44.3 | 28.8 | 59.6 | 43.7 |
| | (+1.8%) | (+2.8%) | (+3.4%) | (+2.5%) | (+1.5%) | (+1.2%) | (+2.3%) | (+2.0%) | (+2.2%) |

MATH dataset, where it is only able to infer valid `EFA`s for $\approx 35\%$ of Precalculus questions. In Sec. 2, we introduce a number of unit tests (i.e., verifiable rewards) that indicate whether a generated `EFA` is valid. Here, we show that we can train models to improve on inferring valid `EFA`s by self-training according to these tests. Specifically, we use a rejection-finetuning approach (Zelikman et al., 2022; Singh et al., 2023; Dong et al., 2023), in which we sample `EFA` candidates from a model and filter according to our rewards to construct a training dataset of correct examples. We begin with the MATH training set (7,500 problems) and sample 10 candidate `EFA`s per problem. Candidates failing any of the reward checks are discarded. The remaining valid examples form a dataset for supervised fine-tuning. This process – sampling, filtering, and retraining – is repeated over 5 iterations (see Appendix F.2 for details).

We report the EFA success rates across iterations in Fig. 4, where we group by difficulty levels (left) and by annotated problem category (right). Success rates steadily improve over training iterations, especially for harder problems. At iteration 0 (before training), we observe that harder problems (e.g., Level 5) are also harder to infer `EFA`s for, with EFA success rates $\approx 17\%$ lower for Level 5 than Level 1 problems. Similarly, certain categories like '*Intermediate Algebra*', '*Counting*' and '*Probability*' are harder to infer `EFA`s for. These domains generally see the most significant increases from training. Between iteration 1 and iteration 5, the Intermediate Algebra's EFA success rate showed the most significant increase, rising from 52.93% to 81.38%, and Geometry improved from 65.71% to 85.71%. Additionally, the pass rate for Level 5 problems increased from 65.95% to 78.73%. These changes indicate substantial improvements in the model's ability to infer `EFA` across these dataset slices. The final model trained for 5 iterations becomes the basis for our `EFAGen` method.

## 3.2 EFAS FAITHFULLY CAPTURE THE REASONING PATTERNS OF SEED PROBLEMS

We expect that valid `EFA`s should be able to capture the reasoning patterns of the seed problem, a property we call *faithfulness*. We measure faithfulness by checking if seeing solutions to problem variants generated from an `EFA` can improve a model's solve rate on the original seed problem. If an `EFA` is faithful, then seeing solutions to problem variants generated from it should improve a model's solve rate on the original seed problem. We select all of problems from MATH-Hard for which Llama3.1-8B-Instruct's pass@5 rate $< 50\%$ and for which `EFAGen` can successfully infer an EFA

Table 3: **EFAGen complements existing synthetic data generation approaches.** Performance comparison across different data scales (1k, 2.5k, 5k) when training models on: NuminaMath synthetic data alone, `EFA`-generated data alone, and both combined. The combined approach typically performs best, with `EFA`-generated data generally outperforming the original synthetic data. The (+%) values show absolute improvements over the NuminaMath Synthetic baseline within each scale. 1st-place is **bold**, 2nd is *italicized*.

| Scale | Data Mix | MATH-500 Performance | | | |
|---|---|---|---|---|---|
| | | **Pass@1** | **Pass@5** | **Pass@10** | **MV Acc** |
| 1k | NuminaMath Synthetic | 20.8 | 45.6 | 56.4 | *38.6* |
| | `EFA` Generated | *24.0* | *48.5* | **58.7** | *38.6* |
| | NuminaMath Synthetic + `EFA` Generated | **24.4** | **48.5** | *58.2* | **40.6** |
| | | (+3.7%) | (+2.9%) | (+1.8%) | (+2.0%) |
| 2.5k | NuminaMath Synthetic | 23.0 | *47.6* | 58.5 | 38.8 |
| | `EFA` Generated | *23.1* | 47.0 | 57.2 | 35.8 |
| | NuminaMath Synthetic + `EFA` Generated | **24.9** | **50.5** | **61.1** | **41.6** |
| | | (+1.9%) | (+2.9%) | (+2.6%) | (+2.8%) |
| 5k | NuminaMath Synthetic | 20.9 | 46.3 | 57.0 | *39.8* |
| | `EFA` Generated | *23.6* | *48.6* | *59.2* | *39.8* |
| | NuminaMath Synthetic + `EFA` Generated | **26.7** | **51.9** | **62.1** | **44.0** |
| | | (+5.8%) | (+5.6%) | (+5.0%) | (+4.2%) |

using the gold solution.[1] For each problem, we sample additional problem variants (we ensure their parameters differ from the seed problem) until Llama3.1-8B-Instruct solves one correctly. We then check if Llama3.1-8B-Instruct can solve the original problem, given the variant and its solution as an in-context example. Results in Table 1 (left) show a 23.07% absolute improvement in pass@1 rate, i.e., `EFA`-generated variants demonstrate faithfulness to the reasoning pattern required for the seed problem.

### 3.3 EFAs Encode Learnable, Shared Structure

We expect that valid `EFAs` should generate problem variants that share common structure. While this is hard to define formally, we can informally measure this by checking the *learnability* of the `EFAs`. An `EFA` is learnable if seeing solutions to problem variants generated from it can improve a model's solve rate on other variants generated from the same `EFA`. We sample 1k `EFAs` inferred from the MATH-Hard test set and generate one new variant per `EFA`, forming a held-out set. For each `EFA`, we also identify one variant that Llama3.1-8B-Instruct solves correctly. We then test Llama3.1-8B-Instruct's performance on the held-out set, with and without access to that solved variant as an in-context example. As shown in Table 1, access to one correctly-solved variant improves the model's pass rate on other variants by 16.65% on average. This provides evidence that the `EFAs` encode learnable, shared structure.

### 3.4 Augmentation: EFAs Are Effective at Expanding Static Math Datasets

While high-quality math datasets exist, these are often expensive to construct. `EFAGen` offers a scalable solution by generating diverse, faithful problem variants through `EFAs`, thereby augmenting existing datasets. To demonstrate this, we fine-tune Llama3.1-8B-Base using `EFA`-generated data derived from the MATH training set. Concretely, we annotate 7,500 training problems with step-by-step reasoning from a teacher model (Llama3.1-8B-Instruct). We ensure that the reasoning is correct by filtering out the reasoning that yields incorrect answers. Then, for each of the 7,500 problems, we construct an `EFA` and sample one problem variant. We compare two training settings. In the first setting, we use only the teacher-labeled seed data. In the second, we augment the seed data by adding

---

[1]Based on the intuition that testing for faithfulness requires an `EFA` (i.e., requires a problem that can be solved in principle) but improving requires a problem that is not solved 100% of the time.

Table 4: **EFAGen can infer EFAs for large-scale competition-level mathematics.** Across 10,000 competition-level problems in NuminaMath, we successfully infer EFAs at substantial rates across different sources. The 95% confidence intervals are significantly above 0% (lowest is 33.7%), demonstrating that EFAGen can reliably infer EFAs for the hardest problems available in large math training datasets.

| Source | Success Rate (%) | 95% CI (%) | Num Problems |
| --- | --- | --- | --- |
| Olympiads | 38.4 | [37.2%, 39.5%] | 6,950 |
| Synthetic AMC | 50.9 | [49.0%, 52.7%] | 2,881 |
| AMC-AIME | 40.6 | [33.7%, 48.4%] | 169 |

EFA-generated examples in a 1:1 ratio. We perform experiments with both 33% (2,500) and 100% (7,500) of the seed data and evaluate performance on three benchmarks: MATH-500 split (Lightman et al., 2023) and the November and December splits of FnEval, each containing perturbed versions of MATH problems. See Appendix F.4 for hyperameter details.

Table 2 shows that EFA-based augmentation leads to consistent improvements across all evaluation metrics: Pass@1, Pass@10 rate, and Majority@25 (Wang et al., 2022), e.g., in the 33% seed setting, Pass@1 improves by +1.9 on MATH-500 and by +2.2 on both FnEval splits. In the 100% seed setting, the gain still holds, underscoring the value of EFAs in enhancing data quality and model performance.

### 3.5 EFAGen Complements Existing Synthetic Data Generation Approaches

EFAs are designed to complement, not replace, existing synthetic data generation approaches. To demonstrate this complementary relationship, we conduct experiments with high-quality synthetic data from NuminaMath (Li et al., 2024), which aggregates synthetic data from various sources, showing that EFAGen can infer EFAs for synthetic data and use these EFAs to augment synthetic datasets at different scales.

We sample 1k, 2.5k, and 5k problems with step-by-step solutions from the synthetic_math and synthetic_amc sources in NuminaMath. For each sample, we apply EFAGen to infer EFAs, generate one problem variant from each EFA, and use rejection sampling to create training data from the EFAs. We train three models at each scale: one trained only on the NuminaMath synthetic data (*NuminaMath Synthetic*), one trained only on data derived from EFAs (*EFA Generated*), and one trained on the NuminaMath synthetic data augmented with our EFA-derived data (*NuminaMath Synthetic + EFA Generated*).

Results on MATH-500 are shown in Table 3. At each scale, the model trained on synthetic data augmented with EFA-generated data performs best across most metrics. Notably, the EFA-generated data typically outperforms the original synthetic NuminaMath data, suggesting that the EFA inference process produces high-quality problem variants that enhance model learning. These results demonstrate that EFAGen provides a scalable approach for augmenting existing synthetic datasets, effectively complementing current synthetic data generation methods.

### 3.6 Generality: EFAGen Can Work Across Diverse Math Domains

Importantly, EFAGen generalizes beyond the distribution of questions in the MATH dataset. As detailed in Fig. 5, our approach successfully infers EFAs across various math sources from the NuminaMath dataset (Li et al., 2024) – ranging from grade-school problems (GSM8K) to national/international competitions (e.g., AMC, AIME, IMO). This demonstrates the broad applicability of EFAs for structuring and scaling math data across diverse domains. We generally see that easier math domains like GSM8K are easier to infer EFAs for than harder domains like AIME or Olympiad problems; nevertheless, EFAGen can infer some successful EFAs even on the hardest domain.

To further demonstrate the scalability of EFAGen, we evaluate its performance on a larger set of 10,000 competition-level math problems from NuminaMath. As shown in Table 4, we are able to successfully infer EFAs at rates of 38.4%, 50.9%, and 40.6% for the Olympiads, Synthetic AMC, and AMC-AIME sources in NuminaMath, respectively. The 95% confidence intervals for each source are significantly above 0% (the lowest is 33.7%), demonstrating that EFAGen can reliably infer EFAs for the hardest problems in large math training datasets.

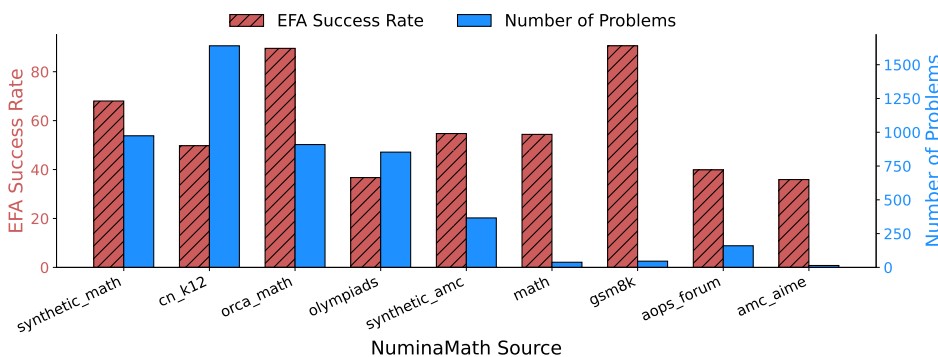

Figure 5: **EFAGen can infer EFAs for diverse sources of math problems.** Here, we show the results of applying `EFAGen` to infer `EFAs` for the NuminaMath (Li et al., 2024) dataset, which contains a mix of math problems from a diversity of sources ranging from grade school mathematics (GSM8K) to national/international olympiads (olympiads). `EFAGen` achieves a nonzero success rate across all sources of problems.

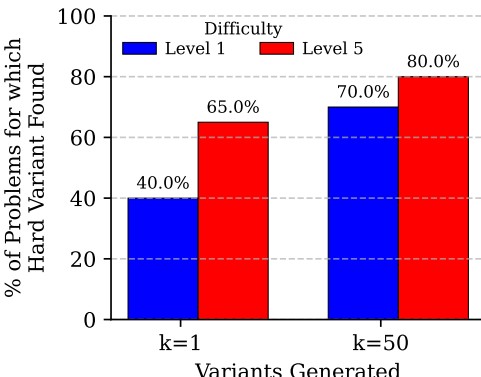

Figure 6: **EFAs can find harder variants of problems.** We infer an `EFA` for a sample of Level 1 (easiest) and Level 5 (hardest) seed problems GPT-4o solves correctly, and generate $k$ variants of each problem. We plot the percentage of seed problems for which a variant that GPT-4o solved incorrectly was found.

### 3.7 ADVERSARIAL SEARCH: EFAGEN CAN FIND HARD PROBLEM VARIANTS

`EFAs` can also be used for evaluation or as a source of targeted training data by finding hard instances that models struggle with.

To demonstrate this, we randomly sample problems from the MATH training that are correctly solved by a strong model (GPT-4o); we sample $N = 20$ of both Level 1 (easiest) and Level 5 (hardest) problems. For each problem, we construct an `EFA` using `EFAGen` and then sample 50 variants from the `EFA`. We attempt to solve each variant with GPT-4o, and measure for what fraction of problems we are able to find variants among the 50 samples that GPT-4o cannot solve. This is an estimate of the probability that we can use an `EFA` to sample problems that cannot be solved by the model, even when the seed problem is solvable. The results are shown in Fig. 6 where we see that there is a non-zero probability of finding hard variants to a given problem, even for easy problems (i.e., Level 1 in MATH) and with a strong model like GPT-4o.

## 4 RELATED WORK

**Symbolic Approaches to Math Reasoning.** A distinct line of prior work has focused on assessing the true mathematical reasoning capabilities of LLMs, specifically by measuring the "reasoning gap" or the drop in math reasoning performance after perturbing questions in existing datasets (Shi et al.,

2023; Zhou et al., 2025; Huang et al., 2025; Ye et al., 2025). One prominent approach is to generate different or difficult math questions conditioned on an existing question but test skills by employing frontier models (Zhang et al., 2024; Patel et al., 2025) or human annotators (Srivastava et al., 2024; Shah et al., 2024; Huang et al., 2025). For instance, Srivastava et al. (2024) propose FnEval dataset by manually functionalizing select problems from the MATH dataset (Hendrycks et al., 2021) that can be subsequently used to sample multiple distinct math problems testing similar skills (albeit with different numerical variables). Similarly, Mirzadeh et al. (2025) release the GSM-Symbolic dataset that augments the existing GSM8K dataset (Cobbe et al., 2021) with templates containing placeholders for several numeric and textual variables and can be used to sample distinct math word problems for a robust evaluation of LLM's reasoning abilities. In contrast, to this line of work requiring expensive annotations from humans or frontier models (thereby, hindering scalability) and tailored to specific, predefined math datasets (c.f. Fig. 2); we propose EFAGen that automatically functionalizes *any* math problem using relatively small language models making it *widely-applicable* and *scalable*, i.e., able to sample a potentially infinite number of related math problems from any distribution or dataset. Moreover, the prior work only focuses on the evaluation of LLMs, whereas we extend the concept of abstraction for downstream applications via training, as shown in Sec. 3.4.

**Data and Environment Generation.** Past work has generally approached improving models on reasoning tasks like math by generating large amounts of broad-coverage training data. This trend builds on work in generating instruction-tuning data (Wang et al., 2023), where model-generated instructions have been used to teach models to follow prompts. Luo et al. (2023) introduced generation method based on Evol-Instruct (Xu et al., 2023), which augmented a seed dataset of math problems by generating easier and harder problems. Related lines of work have sought to expand datasets by augmenting existing math datasets (Yu et al., 2024), adding multiple reasoning strategies (Yue et al., 2024), covering challenging competition problems (Li et al., 2024), or curating responses (Liu et al., 2024). The data generated in these settings differs from our data in a number of respects: first, it is generally broad-coverage, focusing on large-scale diverse data, as opposed to targeted, instance-specific data. This direction was also explored by Khan et al. (2025), who define data generation agents that can generate specific data based on a particular model's weaknesses, covering math and several other domains. Finally, past work that has augmented a seed dataset (e.g., Yu et al. (2024); Yue et al. (2024)) has done so by modifying problems in the surface form, whereas our method first infers a latent structure and then creates problems by sampling from the structure. In contrast, EFAGen focuses on generating similar examples of existing data by inferring an underlying structure from an example; we show that this has applications to data generation for augmentation but also for stress-testing or measuring the performance gap of models on similar problems.

## 5 CONCLUSION

We introduce Executable Functional Abstractions (EFA), a representation of the abstracted logic of a math problem in a parameterized form, enabling the automated sampling of variant problems. We then propose EFAGen, a framework that infers EFAs via program synthesis using large language models (LLMs) that we train using rewards from EFA execution. Our approach over-generate EFA candidates with an LLM and filters them using a suite of property tests that verify their validity. We show that EFAGen successfully infers EFAs for diverse math problems and incorporating execution feedback as a reward in a simple self-training scheme further improves its performance. Models trained on EFA-generated math problems not only perform better on the generated variants but also improve accuracy on the original seed problems. Finally, we show that EFAs provide a scalable solution for augmenting diverse problem variants across various math datasets.

## ETHICS STATEMENT

In this work, we propose an inference-time method, EFAGen that can be used sample additional math problems for training or testing. Consequently, the LLMs utilized by EFAGen may still exhibit stereotypes, biases, and other negative traits inherent in their pre-training data (Weidinger et al., 2021), over which we have no control. Therefore, the outputs produced by EFAGen carry the same potential for misuse as those from other test-time methods. Further research is necessary to assess and mitigate these biases in LLMs. Additionally, care must be taken when executing LLM-generated code which can be erroneous and cause unrecoverable changes to the system files.

## REPRODUCIBILITY STATEMENT

We will open source our code and data to aid replication of our findings. We also provide implementation details of `EFAGen` in Sec. 2 and prompts in Appendix F. The math datasets we use are all publicly available.

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

## A  APPENDIX

The section Adversarial Search (Fig. 6) outlines how EFAs can generate challenging problem variants to probe model weaknesses. The Scaling section (Appendix C) investigates the effect of the number of sampled variants per EFA, showing how performance trends with increased augmentation. The Ablation section (Appendix D) analyzes the impact of applying unit tests during EFA generation on downstream data quality. Qualitative Examples (Appendix E) presents representative EFAs spanning several MATH domains, including algebra, number theory, and probability, illustrating the range and structure captured by the method. The Experimental Details section describes all data generation, augmentation, and model training settings—EFA generation (box F.1), rejection finetuning and variant sampling protocols (Appendix F.2), math inference configuration, and details for math-specific training (Appendix F.4).

## B  QUALITY ANALYSIS: LOW-QUALITY EFAS ARE NATURALLY FILTERED OUT

A potential concern with EFAGen is that the automated EFA generation process may produce low-quality abstractions that could negatively impact training. To address this, we analyze how rejection sampling naturally filters out problematic EFAs during the training data generation process.

We identify "bad" EFAs using an LLM with heuristics that flag abstractions exhibiting common failure modes: trivial problems, extraneous variables, or hard-coded values. We then compare the training data yield rates (the percentage of responses that receive non-zero rewards during rejection sampling) between good and bad EFAs.

As shown in Table 5, low-quality EFAs have significantly lower yield rates compared to good EFAs. With a single answer attempt, bad EFAs contribute training data only 5.04% of the time, compared to 27.0% for good EFAs – a ratio of over 5 to 1 in favor of good data. Even when allowing up to 5 answer attempts, the ratio remains favorable at 4.51 to 1. This demonstrates that as long as rejection

**EFA for Arithmetic Word Problem**

**Original Problem**

When Sophie watches her nephew, she gets out a variety of toys for him. The bag of building blocks has 31 blocks in it. The bin of stuffed animals has 8 stuffed animals inside. The tower of stacking rings has 9 multicolored rings on it.Sophie recently bought a tube of bouncy balls, bringing her total number of toys for her nephew up to 62. How many bouncy balls came in the tube?

```python
class Problem(BaseModel):
    blocks: int
    animals: int
    rings: int
    total: int

    @classmethod
    def original(cls) → Self:
        return cls(blocks=31, animals=8, rings=9, total=62)

    @classmethod
    def sample(cls) → Self:
        blocks = random.randint(20, 50)
        animals = random.randint(5, 15)
        rings = random.randint(5, 15)
        balls = random.randint(10, 40)
        total = blocks + animals + rings + balls
        return cls(blocks=blocks, animals=animals, rings=rings,
total=total)

    def render(self) → str:
        return (
            f"When Sophie watches her nephew, she gets out a variety
of toys for him. "
            f"The bag of building blocks has {self.blocks} blocks in
it. "
            f"The bin of stuffed animals has {self.animals} stuffed
animals inside. "
            f"The tower of stacking rings has {self.rings}
multicolored rings on it."
            "Sophie recently bought a tube of bouncy balls, bringing
her total number of toys "
            f"for her nephew up to {self.total}. How many bouncy
balls came in the tube?"
        )

    def solve(self) → str:
        known = self.blocks + self.animals + self.rings
        balls = self.total - known
        return str(balls)
```

**EFA for Tridiagonal Matrix Eigenvalue Problem**

**Original Problem**

Suppose you are given the matrix:

$$A = \begin{bmatrix} 2 & 1 & 0 \\ 1 & 3 & 1 \\ 0 & 1 & 4 \end{bmatrix}$$

Find the eigenvalues of the matrix.

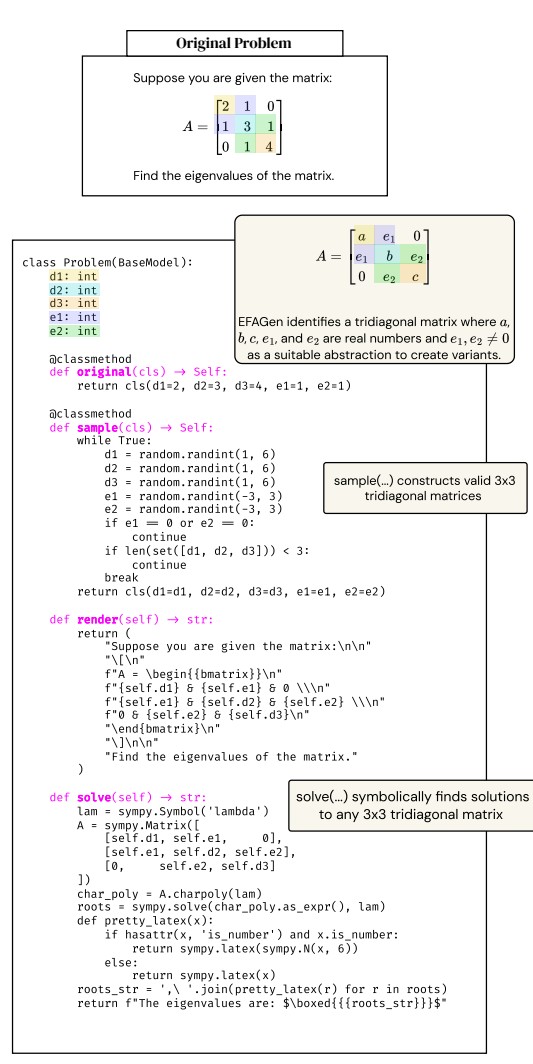

$$A = \begin{bmatrix} a & e_1 & 0 \\ e_1 & b & e_2 \\ 0 & e_2 & c \end{bmatrix}$$

EFAGen identifies a tridiagonal matrix where $a$, $b$, $c$, $e_1$, and $e_2$ are real numbers and $e_1, e_2 \neq 0$ as a suitable abstraction to create variants.

```python
class Problem(BaseModel):
    d1: int
    d2: int
    d3: int
    e1: int
    e2: int

    @classmethod
    def original(cls) → Self:
        return cls(d1=2, d2=3, d3=4, e1=1, e2=1)

    @classmethod
    def sample(cls) → Self:
        while True:
            d1 = random.randint(1, 6)
            d2 = random.randint(1, 6)
            d3 = random.randint(1, 6)
            e1 = random.randint(-3, 3)
            e2 = random.randint(-3, 3)
            if e1 == 0 or e2 == 0:
                continue
            if len(set([d1, d2, d3])) < 3:
                continue
            break
        return cls(d1=d1, d2=d2, d3=d3, e1=e1, e2=e2)

    def render(self) → str:
        return (
            "Suppose you are given the matrix:\n\n"
            "\[\n"
            f"A = \begin{{bmatrix}}\n"
            f"{self.d1} & {self.e1} & 0 \\\n"
            f"{self.e1} & {self.d2} & {self.e2} \\\n"
            f"0 & {self.e2} & {self.d3}\n"
            "\end{bmatrix}\n"
            "\]\n\n"
            "Find the eigenvalues of the matrix."
        )

    def solve(self) → str:
        lam = sympy.Symbol('lambda')
        A = sympy.Matrix([
            [self.d1, self.e1,      0],
            [self.e1, self.d2, self.e2],
            [0,      self.e2, self.d3]
        ])
        char_poly = A.charpoly(lam)
        roots = sympy.solve(char_poly.as_expr(), lam)
        def pretty_latex(x):
            if hasattr(x, 'is_number') and x.is_number:
                return sympy.latex(sympy.N(x, 6))
            else:
                return sympy.latex(x)
        roots_str = ',\ '.join(pretty_latex(r) for r in roots)
        return f"The eigenvalues are: $\boxed{{{roots_str}}}$"
```

sample(…) constructs valid 3x3 tridiagonal matrices

solve(…) symbolically finds solutions to any 3x3 tridiagonal matrix

Figure 7: `EFA`s inferred for problems shown in Fig. 2. On the left is an `EFA` for a grade-school level math word problem. On the right is an `EFA` for the tridiagonal matrix eigenvalue problem. `EFA`s are able to represent both types of problems, despite the wide gap in problem complexity. The `sample` method constructs mathematical objects with required properties, while the `solve` method implements a generalized solution for any object constructible by the `sample` method. See Sec. 2.2 for a more detailed explanation.

Table 5: **Low-quality `EFA`s are naturally filtered out during rejection sampling.** We compare the training data yield rates (percentage of responses that receive non-zero rewards) between good and bad `EFA`s. Bad `EFA`s are identified using LLM-based heuristics that flag trivial problems, extraneous variables, or hard-coded values. The low yield rates of bad `EFA`s mean they contribute minimally to training data.

|  | Good `EFA`s | Bad `EFA`s | Good to Bad Data Ratio |
|---|---|---|---|
| Training Data Yield Rate (1 Answer Attempt) | 27.0% | 5.04% | 5.36 to 1 |
| Training Data Yield Rate (5 Answer Attempts) | 39.9% | 8.85% | 4.51 to 1 |

Table 6: **`EFA`-generated data performs comparably to real data.** Direct comparison of training exclusively on problem variants generated by `EFA`s versus training exclusively on real problems from the MATH training set. Despite potential noise in rejection-sampled `EFA` data, models trained on synthetic data achieve nearly identical performance to those trained on real data.

| Training Data | MATH-500 | | | FnEval (November) | | | FnEval (December) | | |
|---|---|---|---|---|---|---|---|---|---|
|  | Pass @ 1 | Pass @ 10 | Maj @ 25 | Pass @ 1 | Pass @ 10 | Maj @ 25 | Pass @ 1 | Pass @ 10 | Maj @ 25 |
| Real Data Only | 22.4 | 56.4 | 36.8 | 24.4 | 55.4 | 39.3 | 24.5 | 55.3 | 39.6 |
| Synthetic Data Only | 22.6 | 58.0 | 37.8 | 24.9 | 56.6 | 38.3 | 25.5 | 57.2 | 40.0 |

sampling or reinforcement learning is used, noisy `EFA`s naturally filter themselves out, ensuring that good data significantly outnumbers bad data in the final training set.

To further validate the quality of `EFA`-generated data, we conduct a direct comparison between training exclusively on problem variants generated by `EFA`s versus training exclusively on real problems from the MATH training set. As shown in Table 6, despite potential noise in rejection-sampled `EFA` data, models trained on synthetic data achieve nearly identical performance to those trained on real data (22.6% vs 22.4% Pass@1 on MATH-500). This shows that `EFA`-generated data is as effective as existing math data for model training.

## C  SCALING: EFAGEN SCALES EFFECTIVELY UP TO 16 EXAMPLES PER EFA

To understand the scaling behavior of `EFA`-based data augmentation, we investigate how performance varies with the number of problem variants generated per `EFA`. We sample 100 unique `EFA`s from the MATH training set and vary the number of problem variants generated by each `EFA` from 1 to 64. For each scaling setting, we train Llama3.1-8B-Base on the generated data and evaluate on MATH-500.

As shown in Table 7, we observe smooth scaling improvements as we increase the number of variants from 1 to 16 examples per `EFA`, with performance gains plateauing beyond 16 examples. Specifically, Pass@1 improves from 14.1% with 1 example per `EFA` to 23.8% with 16 examples, while Pass@10 increases from 48.5% to 57.6% over the same range. However, scaling begins to saturate at 32 and 64 examples per `EFA`, suggesting that sampling too many problem variants from each `EFA` uniformly may hurt diversity and lead to diminishing returns. The optimal scaling point appears to be around 16 examples per `EFA`, where three of the four metrics achieve their peak performance.

## D  ABLATION: UNIT TESTS IMPROVE EFA-BASED DATA AUGMENTATION QUALITY

Despite some errors in `EFA` generation, we find that the current `EFA`s are effectively improving performance. When we lower the quality by removing our unit tests, the performance gains from augmentation also decrease. As shown in Table 8, applying unit tests consistently improves performance across all benchmarks and metrics. The unit tests provide an average improvement of 2.2 percentage points on MATH-500 Pass@1, 1.7 percentage points on FnEval November Pass@1, and 2.9 percentage points on FnEval December Pass@1.

In general, we believe there is a tradeoff between the level of noise in generated data and the cost of data generation, and `EFA`s occupy a generally useful point on the tradeoff curve. We can change the

Table 7: **EFAGen scales effectively up to 16 examples per EFA.** We train Llama3.1-8B-Base on varying numbers of problem variants generated from each `EFA` and evaluate on MATH-500. Performance improves smoothly from 1 to 16 examples per `EFA`, with diminishing returns beyond that point. Bold numbers indicate the best performance for each metric.

| Training Data per EFA | Pass@1 | Pass@5 | Pass@10 | Majority Vote Accuracy |
|---|---|---|---|---|
| 1 | 14.1 | 37.2 | 48.5 | 29.6 |
| 2 | 19.1 | 42.8 | 53.3 | 34.0 |
| 4 | 21.9 | 45.1 | 54.7 | 35.4 |
| 8 | 22.9 | 46.9 | 57.4 | 35.6 |
| 16 | 23.8 | **47.6** | **57.6** | **37.4** |
| 32 | **24.3** | 46.6 | 56.4 | 37.2 |
| 64 | 23.9 | 45.6 | 55.2 | 36.2 |

Table 8: **Unit tests improve `EFA`-based data augmentation quality.** We compare the performance of `EFA`-based data augmentation with and without the unit tests that filter out low-quality `EFA`s. The unit tests consistently improve performance across all benchmarks, demonstrating their effectiveness in maintaining data quality.

| | MATH-500 | | | FnEval (November) | | | FnEval (December) | | |
|---|---|---|---|---|---|---|---|---|---|
| Unit Tests | Pass @ 1 | Pass @ 10 | Maj @ 25 | Pass @ 1 | Pass @ 10 | Maj @ 25 | Pass @ 1 | Pass @ 10 | Maj @ 25 |
| False | 20.4 | 55.2 | 35.4 | 24.1 | 54.7 | 35.7 | 22.0 | 55.2 | 37.1 |
| True | **22.6** | **58.0** | **37.8** | **24.9** | **56.6** | **38.3** | **25.5** | **57.2** | **40.0** |
| $\Delta$ | (+2.2%) | (+2.8%) | (+2.4%) | (+0.8%) | (+1.9%) | (+2.6%) | (+3.5%) | (+2.0%) | (+2.9%) |

tradeoff and reduce noise by increasing the cost of filtering and data generation. These same issues occur with synthetic data generation approaches. The value of our approach is that data generation can be replaced with program execution rather than a call to a frontier LLM.

# E    QUALITATIVE EXAMPLES

In this section, we display qualitative examples of `EFA`s across the MATH training set which were validated by our tests.

---

**Box  E.1| `EFA` (Algebra)**

**Original Problem**
Solve the equation:

$$\sqrt{5x-4} + \frac{15}{\sqrt{5x-4}} = 8$$

**Original Solution**
Let $u = \sqrt{5x-4}$. We can rewrite the equation in terms of $u$ :

$$u + \frac{15}{u} = 8$$

Multiply through by $u$ to remove the fraction:

$$u^2 + 15 = 8u$$

Reorganize into a standard quadratic form:

$$u^2 - 8u + 15 = 0$$

Factorize the quadratic:

$$(u-5)(u-3) = 0$$

---

Thus, $u = 5$ or $u = 3$. For $u = 5$:

$$\sqrt{5x - 4} = 5 \Rightarrow 5x - 4 = 25 \Rightarrow 5x = 29 \Rightarrow x = \frac{29}{5}$$

For $u = 3$:

$$\sqrt{5x - 4} = 3 \Rightarrow 5x - 4 = 9 \Rightarrow 5x = 13 \Rightarrow x = \frac{13}{5}$$

Therefore, the solutions are $x = \frac{29}{5}, \frac{13}{5}$. Conclusion: The possible values of $x$ are $\frac{29}{5}$ and $\frac{13}{5}$, which satisfy the original equation when we substitute back, confirming the correctness of our solution.

**Inferred `EFA`**

```python
class Problem(BaseModel):
    coefficient: int
    constant: int
    term: int

    @classmethod
    def original(cls) ->Self:
        return cls(coefficient=5, constant=4, term=15)

    @classmethod
    def sample(cls) ->Self:
        coefficient = random.randint(3, 10)
        constant = random.randint(1, 5)
        term = random.randint(10, 20)
        return cls(coefficient=coefficient, constant=constant, term=
            term)

    def render(self) ->str:
        return f"""Solve the equation:
\\[ \\sqrt{{{self.coefficient}x-{self.constant}}} + \\frac{{{self.
    term}}}{{\\sqrt{{{self.coefficient}x-{self.constant}}}}} = 8.
    \\]"""

    def solve(self) ->str:
        u = sympy.symbols('u')
        equation = sympy.Eq(u + self.term / u, 8)
        u_values = sympy.solve(equation, u)
        x_solutions = []
        for u_val in u_values:
            x_equation = sympy.Eq(u_val ** 2, self.coefficient * sympy.

                symbols('x') - self.constant)
            x_solution = sympy.solve(x_equation, sympy.symbols('x'))
            x_solutions.extend(x_solution)
        return ', '.join(
            f'\\frac{{{sol.as_numer_denom()[0]}}}{{{sol.as_numer_denom
                ()[1]}}}'
             for sol in x_solutions)
```

**Variant generated by `EFA`**

Solve the equation:

$$\sqrt{7x - 5} + \frac{14}{\sqrt{7x - 5}} = 8$$

*Solution*:

$$\frac{23 - 8\sqrt{2}}{7}, \frac{8\sqrt{2} + 23}{7}$$

---

**Box E.2| `EFA` (Number Theory)**

**Original Problem**

How many positive divisors does $8!$ have?

**Original Solution**

First, calculate $8!$ : $8! = 1 \times 2 \times 3 \times 4 \times 5 \times 6 \times 7 \times 8 = 40320$. Next, find the prime factorization of $40320 : 40320 = 2^7 \times 3^2 \times 5^1 \times 7^1$. Now, apply the formula for counting the divisors: If $n = p^a \times q^b \times r^c \times \ldots$, then the number of divisors $t(n)$ is given by:

$$t(n) = (a+1)(b+1)(c+1)\ldots$$

Here $a = 7, b = 2, c = 1, d = 1$ for the primes $2, 3, 5$, and $7$ respectively. Applying the formula:

$$t(40320) = (7+1)(2+1)(1+1)(1+1) = 8 \times 3 \times 2 \times 2 = 96$$

Conclusion: The result is consistent with the factorial and prime factorization, providing a logically correct count of divisors.

**Inferred `EFA`**

```
class Problem(BaseModel):
    n: int

    @classmethod
    def original(cls) ->Self:
        return cls(n=8)

    @classmethod
    def sample(cls) ->Self:
        n = random.randint(4, 10)
        return cls(n=n)

    def render(self) ->str:
        return f'How many positive divisors does {self.n}! have?'

    def solve(self) ->str:
        factorial_value = math.factorial(self.n)
        factors = sympy.factorint(factorial_value)
        divisor_count = 1
        for exponent in factors.values():
            divisor_count *= exponent + 1
        return str(divisor_count)
```

**Variant generated by `EFA`**

How many positive divisors does $9!$ have?

*Solution*:

$$160$$

---

**Box E.3| `EFA` (Probability)**

**Original Problem**

Two 8-sided dice are tossed. What is the probability that the sum of the numbers shown on the dice is a prime number? Express your answer as a common fraction.

**Original Solution**

Let $d_1$ and $d_2$ be the outcomes of the two 8-sided dice, where $d_1, d_2 \in \{1, 2, \ldots, 8\}$.
The total number of possible outcomes in the sample space is:

$$|\Omega| = 8 \times 8 = 64$$

We want to find the number of outcomes where the sum $S = d_1 + d_2$ is a prime number. The smallest possible sum is $1 + 1 = 2$ and the largest is $8 + 8 = 16$. The prime numbers in this range are $2, 3, 5, 7, 11$, and $13$.

We enumerate the pairs $(d_1, d_2)$ corresponding to each prime sum:

**Sum = 2:** $\{(1,1)\}$   (1 way)

**Sum = 3:** $\{(1,2),(2,1)\}$   (2 ways)

**Sum = 5:** $\{(1,4),(4,1),(2,3),(3,2)\}$   (4 ways)

**Sum = 7:** $\{(1,6),(6,1),(2,5),(5,2),(3,4),(4,3)\}$   (6 ways)

**Sum = 11:** $\{(3,8),(8,3),(4,7),(7,4),(5,6),(6,5)\}$   (6 ways)

**Sum = 13:** $\{(5,8),(8,5),(6,7),(7,6)\}$   (4 ways)

The total number of favorable outcomes is the sum of the ways for each prime:
$$N_{\text{favorable}} = 1 + 2 + 4 + 6 + 6 + 4 = 23$$

The probability is the ratio of the number of favorable outcomes to the total number of outcomes:
$$P(\text{Sum is prime}) = \frac{N_{\text{favorable}}}{|\Omega|} = \frac{23}{64}$$

The final answer is:
$$\boxed{\frac{23}{64}}$$

**Inferred EFA**

```python
class Problem(BaseModel):
    sides: int

    @classmethod
    def original(cls) ->Self:
        return cls(sides=8)

    @classmethod
    def sample(cls) ->Self:
        sides = random.randint(4, 20)
        return cls(sides=sides)

    def solve(self) ->str:
        total_outcomes = self.sides * self.sides

        def is_prime(n: int) ->bool:
            if n <= 1:
                return False
            for i in range(2, int(math.sqrt(n)) + 1):
                if n % i == 0:
                    return False
            return True
        primal_sum_occurrences = 0
        for die1 in range(1, self.sides + 1):
            for die2 in range(1, self.sides + 1):
                sum_of_dice = die1 + die2
                if is_prime(sum_of_dice):
                    primal_sum_occurrences += 1
        probability = primal_sum_occurrences / total_outcomes
        fraction = sympy.Rational(primal_sum_occurrences,
            total_outcomes)
        return f'\\frac{{{fraction.numerator}}}{{{fraction.
            denominator}}}'

    def render(self) ->str:
        return (
            f'Two {self.sides}-sided dice are tossed. What is the
                probability that the sum of the numbers shown on the
                dice is a prime number? Express your answer as a common
                fraction.'
```

```
            )
```

**Variant generated by `EFAGen`** Two 19-sided dice are tossed. What is the probability that the sum of the numbers shown on the dice is a prime number? Express your answer as a common fraction.
*Solution*:
$$\frac{105}{361}$$

## F  EXPERIMENTAL DETAILS

### F.1  GENERATING EFAS

When generating EFAs, we use the prompt in box F.1. To sample multiple candidates for EFAs, we use beam search with a temperature of 0.7 and a max generation length of 4096. We extract the resulting EFAs from the LLMs response by looking for a markdown code block and extracting all markdown code blocks that have the necessary class structure.

---

**Box F.1 | Prompt for Inferring EFAs**

```
# Instructions for Math Problem Functionalization

Your task is to convert a mathematical problem and its solution
    into a reusable Python class that can generate similar problems.
     Follow these steps:

1. Create a Python class that inherits from BaseModel with
    parameters that can vary in the problem. These parameters should
     capture the core numerical or mathematical values that could be
     changed while maintaining the same problem structure.

2. Implement the following required methods:
   - `original()`: A class method that returns the original problem'
       s parameters
   - `sample()`: A class method that generates valid random
       parameters for a similar problem
   - `render()`: An instance method that produces the problem
       statement as a formatted string
   - `solve()`: An instance method that computes and returns the
       solution

3. For the `sample()` method:
   - Generate random parameters that maintain the problem's
       mathematical validity
   - Include appropriate constraints and relationships between
       parameters
   - Use reasonable ranges for the random values

4. For the `render()` method:
   - Format the problem statement using f-strings
   - Include proper mathematical notation using LaTeX syntax where
       appropriate
   - Maintain the same structure as the original problem

5. For the `solve()` method:
   - Implement the solution logic using the instance parameters
   - Return the final answer in the expected format (string,
       typically)
   - Include any necessary helper functions within the method
```

```
6. Consider edge cases and validity:
   - Ensure generated problems are mathematically sound
   - Handle special cases appropriately
   - Maintain reasonable complexity in generated problems

7. Do not import any libraries! The following libraries have been
   imported. Use fully qualified names for all imports:
   - pydantic.BaseModel is imported as `BaseModel`
   - random is imported as `random`
   - math is imported as `math`
   - numpy is imported as `np`
   - sympy is imported as `sympy`
   - typing.Self is imported as `Self`

Example usage:
```python
problem = MyMathProblem.original() # Get original problem
variant = MyMathProblem.sample() # Generate new variant
question = variant.render() # Get problem statement
answer = variant.solve()  # Compute solution
```

The goal is to create a class that can both reproduce the original
    problem and generate mathematically valid variations of the same
     problem type.

# Example 1
## Problem Statement
Evaluate $i^5+i^{-25}+i^{45}$.

## Solution
We have $i^5 = i^4\\cdot i = 1\\cdot (i) = i$. We also have $i
    ^{-25} = 1/i^{25} = 1/(i^{24}\\cdot i) = 1/[1\\cdot (i)] = 1/i =
    \\frac1{i}\\cdot\\frac{i}{i} = i/(-1) = -i$ and $i^{45} =
(i^{44})\\cdot i= 1\\cdot i =i$, and . So, adding these three
    results gives $i^5 + i^{-25} + i^{45} = i+-i+i = \\boxed{i}$.\
    nFinal Answer: The final answer is $\\boxed{ i }$.

## Functionalization
```python
class Problem(BaseModel):
    exponent1: int
    exponent2: int
    exponent3: int

    @classmethod
    def original(cls) -> Self:
        return cls(exponent1=5, exponent2=-25, exponent3=45)

    @classmethod
    def sample(cls) -> Self:
        exponent1 = random.randint(-100, 100)
        exponent2 = random.randint(-100, 100)
        exponent3 = random.randint(-100, 100)
        return cls(exponent1=exponent1, exponent2=exponent2,
            exponent3=exponent3)

    def render(self) -> str:
        return f"Evaluate $i^{{{self.exponent1}}} + i^{{{self.
            exponent2}}} + i^{{{self.exponent3}}}$."

    def solve(self) -> str:
        # Compute the values of i^n mod 4 cycle
```

```
         def compute_i_power(exp: int) -> complex:
             cycle = [1, 1j, -1, -1j] # 1, i, -1, -i
             return cycle[exp % 4]

         # Compute each term
         term1 = compute_i_power(self.exponent1)
         term2 = compute_i_power(self.exponent2)
         term3 = compute_i_power(self.exponent3)

         # Calculate the sum
         result = term1 + term2 + term3

         # Express as LaTeX
         result_latex = (
             f"{result:.0f}" if result.imag == 0 else str(result).
                 replace("j", "i")
         )
         return f"{result_latex}"
```

# Example 2
## Problem Statement
Altitudes $\overline{AX}$ and $\overline{BY}$ of acute triangle
    $ABC$ intersect at $H$. If $\angle BAC = 43^\circ$ and $\angle
    ABC = 67^\circ$, then what is $\angle HCA$?
## Solution
First, we build a diagram:

```
size(150); defaultpen(linewidth(0.8));
pair B = (0,0), C = (3,0), A = (1.2,2), P = foot(A,B,C), Q = foot(B,
    A,C),H = intersectionpoint(B--Q,A--P);
draw(A--B--C--cycle);
draw(A--P^^B--Q);
pair Z;
Z = foot(C,A,B);
draw(C--Z);
label("$A$",A,N); label("$B$",B,W); label("$C$",C,E); label("$X$",P,
    S); label("$Y$",Q,E); label("$H$",H+(0,-0.17),SW);
label("$Z$",Z,NW);
draw(rightanglemark(B,Z,H,3.5));
draw(rightanglemark(C,P,H,3.5));
draw(rightanglemark(H,Q,C,3.5));
```

Since altitudes $\overline{AX}$ and $\overline{BY}$ intersect at
    $H$, point $H$ is the orthocenter of $\triangle ABC$. Therefore,
     the line through $C$ and $H$ is perpendicular to
side $\overline{AB}$, as shown. Therefore, we have $\angle HCA = \
    angle ZCA = 90^\circ - 43^\circ = \boxed{47^\circ}$.

## Functionalization
```python
class Problem(BaseModel):
    angle_BAC: int # angle BAC in degrees
    angle_ABC: int # angle ABC in degrees

    @classmethod
    def original(cls) -> Self:
        return cls(angle_BAC=43, angle_ABC=67)

    @classmethod
    def sample(cls) -> Self:
```

```
        # Generate random acute angles that form a valid triangle
        # Sum of angles must be less than 180
        angle1 = random.randint(30, 75) # Keep angles acute
        angle2 = random.randint(30, 75)
        # Ensure the third angle is also acute
        if angle1 + angle2 >= 150:
            angle1 = min(angle1, 60)
            angle2 = min(angle2, 60)
        return cls(angle_BAC=angle1, angle_ABC=angle2)

    def solve(self) -> str:
        # The angle HCA is complementary to angle BAC
        # This is because H is the orthocenter and CH is
            perpendicular to AB
        angle_HCA = 90 - self.angle_BAC
        return f"{angle_HCA}"

    def render(self) -> str:
        return (
            f"Altitudes $\\overline{{AX}}$ and $\\overline{{BY}}$ of
                acute triangle $ABC$ "
            f"intersect at $H$. If $\\angle BAC = {self.angle_BAC}^\\
                circ$ and "
            f"$\\angle ABC = {self.angle_ABC}^\\circ$, then what is $\\
                angle HCA$?"
        )
```

# Example 3
## Problem Statement
On a true-false test of 100 items, every question that is a
    multiple of 4 is true, and all others are false. If a student
    marks every item that is a multiple of 3 false and all others
    true, how many of the 100 items will be correctly answered?
## Solution
The student will answer a question correctly if

Case 1: both the student and the answer key say it is true. This
    happens when the answer is NOT a multiple of 3 but IS a multiple
     of 4.

Case 2. both the student and the answer key say it is false. This
    happens when the answer IS a multiple of 3 but is NOT a multiple
     of 4.

Since the LCM of 3 and 4 is 12, the divisibility of numbers (in our
     case, correctness of answers) will repeat in cycles of 12. In
    the first 12 integers, $4$ and $8$ satisfy Case 1
and $3,6,$ and $9$ satisfy Case 2, so for every group of 12, the
    student will get 5 right answers. Since there are 8 full groups
    of 12 in 100, the student will answer at least $8
\cdot 5 = 40$ questions correctly. However, remember that we must
    also consider the leftover numbers 97, 98, 99, 100 and out of
    these, $99$ and $100$ satisfy one of the cases. So
our final number of correct answers is $40 + 2 = \boxed{42}$.

## Functionalization
```python
class Problem(BaseModel):
    total_questions: int # Total number of questions
    multiple1: int # First multiple (4 in original problem)
    multiple2: int # Second multiple (3 in original problem)
```

```
        @classmethod
        def original(cls) -> Self:
            return cls(total_questions=100, multiple1=4, multiple2=3)

        @classmethod
        def sample(cls) -> Self:
            # Generate reasonable random parameters
            total = random.randint(50, 200) # Reasonable test length
            # Choose coprimes or numbers with small LCM for interesting
                results
            mult1 = random.randint(2, 6)
            mult2 = random.randint(2, 6)
            while mult1 == mult2: # Ensure different numbers
                mult2 = random.randint(2, 6)
            return cls(total_questions=total, multiple1=mult1, multiple2=
                mult2)

        def solve(self) -> str:
            def lcm(a: int, b: int) -> int:
                def gcd(x: int, y: int) -> int:
                    while y:
                        x, y = y, x % y
                    return x

                return abs(a * b) // gcd(a, b)

            # Find cycle length (LCM)
            cycle_length = lcm(self.multiple1, self.multiple2)

            # Count correct answers in one cycle
            correct_per_cycle = 0
            for i in range(1, cycle_length + 1):
                answer_key_true = i % self.multiple1 == 0
                student_true = i % self.multiple2 != 0
                if answer_key_true == student_true:
                    correct_per_cycle += 1

            # Calculate complete cycles and remainder
            complete_cycles = self.total_questions // cycle_length
            remainder = self.total_questions % cycle_length

            # Calculate total correct answers
            total_correct = complete_cycles * correct_per_cycle

            # Add correct answers from remainder
            for i in range(1, remainder + 1):
                answer_key_true = i % self.multiple1 == 0
                student_true = i % self.multiple2 != 0
                if answer_key_true == student_true:
                    total_correct += 1

            return str(total_correct)

        def render(self) -> str:
            return (
                f"On a true-false test of {self.total_questions} items, "
                f"every question that is a multiple of {self.multiple1} is
                    true, "
                f"and all others are false. If a student marks every item
                    that is "
                f"a multiple of {self.multiple2} false and all others true,
                    how "
```

```
            f"many of the {self.total_questions} items will be
                correctly answered?"
        )
    ```

    # Your Turn
    Functionalize the following problem:

    ## Problem Statement
    [% problem_statement %]

    ## Solution
    [% solution %]

    ## Functionalization
```

## F.2 EFAGEN TRAINING DETAILS

When doing rejection finetuning, we sample 20 candidate EFAs programs from the LLM for each seed problem during the rejection sampling phase. We sample 20 variants from each EFA in order to run the **has_dof(EFA)** and **is_single_valued(EFA)** tests. When finetuning on the EFAs that pass all tests, we use the the same prompt box F.1 as the instruction and the extracted code of the EFA as the response. We use Transformers (Wolf et al., 2020) and Llama-Factory (Zheng et al., 2024) libraries for training. We format all data in the Alpaca format (Taori et al., 2023) as instruction-response pairs. We use the Adam optimizer with a batch size of 16 and a cosine learning rate scheduler with a warmup ratio of 0.1 and train for 3 epochs in the FP16 datatype. We apply LoRA to all linear layers with a rank of 16 and an alpha of 32, no bias, and a dropout of 0.05. We truncate all training examples to a maximum length of 4096 tokens with a batch size of 32.

## F.3 MATH INFERENCE SETTINGS

When doing 0-shot inference with Llama3.1-8B-Instruct, we use the official Llama3.1 prompt in box F.2. When doing few-shot inference with Llama3.1-8B-Instruct, we use a modified version of the official prompt, shown in box F.3. When sampling multiple responses, we use beam search with a temperature of 0.7 and a max generation length of 2048. When sampling a single response, we use beam search with a temperature of 0.0 and a max generation length of 2048. In all cases, we check for equality of answers using the math-verify library.

---

**Box F.2| Llama3.1 0-shot MATH Prompt**

```
Solve the following math problem efficiently and clearly:

- For simple problems (2 steps or fewer):
Provide a concise solution with minimal explanation.

- For complex problems (3 steps or more):
Use this step-by-step format:

## Step 1: [Concise description]
[Brief explanation and calculations]

## Step 2: [Concise description]
[Brief explanation and calculations]

...

Regardless of the approach, always conclude with:
```

```
Therefore, the final answer is: $\boxed{answer}$. I hope it is
    correct.

Where [answer] is just the final number or expression that solves
    the problem.

Problem: {{ instruction }}
```

**Box F.3| Llama3.1 N-shot MATH Prompt**

```
Solve the following math problem efficiently and clearly:

- For simple problems (2 steps or fewer):
Provide a concise solution with minimal explanation.

- For complex problems (3 steps or more):
Use this step-by-step format:

\#\# Step 1: [Concise description]
[Brief explanation and calculations]

\#\# Step 2: [Concise description]
[Brief explanation and calculations]

...

Regardless of the approach, always conclude with:

Therefore, the final answer is: $\boxed{answer}$. I hope it is
    correct.

Where [answer] is just the final number or expression that solves
    the problem.

Here are some examples:
{% for few_shot_example in few_shot_examples %}
Problem: {{ few_shot_example.instruction }}
{{ few_shot_example.response }}
{% endfor %}

Problem: {{ instruction }}
```

## F.4 MATH TRAINING DETAILS

We use the same hyperparameters and chat data format as in Appendix F.2, except we cutoff training data over 2048 tokens. However, we use a simpler prompt template, shown in box F.4 to format the teacher responses. When annotating with a Llama3.1-8B-Instruct teacher, we sample 5 responses per math problem with a temperature of 0.7. We check for equality of answers using the math-verify library.

**Box F.4| Minimal instruction-tuning prompt used for augmentation experiments**

```
Question: {{ question }}
Step-by-step Answer
```

