# OpenReview forum: "Executable Functional Abstractions: Inferring Generative Programs for Advanced Math Problems"
_ICLR.cc/2026/Conference — Submitted to ICLR 2026_

### Official Review · Reviewer_eXQi · 2025-10-31

**Soundness:** 3
**Presentation:** 2
**Contribution:** 2
**Rating:** 2
**Confidence:** 3

**Summary:**

This paper proposes a new form of synthetic data generation for mathematical reasoning based on EFA's (Executable Functional Abstractions). An EFA is a parameterized abstraction of a math problem that can generate new instances of a problem with the same structure but different input numbers and a different solution number. First, the authors propose a process for generating EFA's with an LLM. They find that conducting iterative fine-tuning on successfully generated EFA's improves a base model's ability to generate EFAs. They show that prompting a LLM to solve a problem given a solution to an EFA-derived example improves performance compared to solving the problem alone. They last show that fine tuning on EFA-derived problems improves performance on MATH-500 and FnEval, whether on its own or combined with other synthetic generation.

**Strengths:**

- EFA's are a principled way to augment data for mathematical problems.
- The authors conduct an extensive variety of experiments on generating EFA's and improving model performance using EFA's
- Overall, the experiments are moderately well executed, although I have several questions about choices for fair comparison
- Using an EFA for synthetic data generation is a general approach that could apply to a lot of domains
- Given an existing solution, one can check EFA's for correctness with reasonable certainty, so it is a relatively safe form of data augmentation / synthetic data generation
- In some sense, fine-tuning on EFA's can be thought of as one form of having the model "self-train" on new instances of the same problem, which is a significant and original idea.

**Weaknesses:**

I would put the weaknesses of this paper in three main categories:

W1. Limited novelty and impact of the approach and results

- Once you understand the definition of an EFA, their usefulness, implementation, and how to discover them is quite simple. However, the authors do a drawn-out presentation of the definition, desiderata, implementation, and verification of EFA's in section 2. To me, this is not much of a contribution, and the presentation is drawn out. For example, section 2.2 is entirely about how to implement EFA's in Python, which is too simple to be in the main text, and probably can be left out of the paper entirely.

- The authors originally present the task of inferring EFA's from a concrete problem as a challenge, but then use simple methods to do so: prompt a LLM to generate an EFA given a problem and its solution, then check for correctness.

- The authors show that showing a solution to an EFA-derived similar problem in context to a problem improves a model's performance, but this is not surprising to me, given how similar EFA-derived problems are.

- The main benefit of EFA's would be in a data augmentation scheme (sections 3.4 and 3.5). However, the performance improvement from EFA augmentation is quite small (~2-3% on average). Moreover, the increase in performance is only shown for a single 8B model, and it's unclear whether the method will help more capable models.

W2. Some evaluations could be more fair

- For Table 1, the data generation process could bias the results in favor of EFA's. To start, only problems are used with pass rate < 50% and for which EFAGen can successfully infer an EFA. This is the type of problem that using an EFA would help the most. This is okay for saying an effect exists, but this overstates the overall average benefit of using EFA's, which the percent improvement numbers could be construed to mean.

- For section 3.4 and Table 2, a potential confounder of the benefit of EFA's is that the EFA-models are trained with more data. What if you control the number of total examples the model is fine-tuned on? For example, instead of 7500 teacher problems and 7500 EFA problems, you could compare 7500 teacher problems with 3750 teacher problems and 3750 EFA problems. This seems like a fair comparison to include.

- Similarly, for section 3.5 and Table 3, different methods (rows) might have a different number of total training examples. How does the number of examples for each of the data mixes vary, and what happens if you control the number of examples to be the same?


W3. Writing quality and presentation

- The paper overelaborates on much of the material, making it long and fluff. For example, section 2.2 is entirely about how to implement EFA's in Python, which is too simple to be in the main text, and probably can be left out of the paper entirely. Other sections draw out what could be simple explanations, perhaps to make the contribution seem larger.

- The abstract is unusually long. The introduction is surprisingly long too, taking over 3 pages of the paper. This is more than the main experiments which take up less than 3 pages.

- There is a lack of precision in much of the writing, and many claims are overexaggerated. For example, in the introduction, the authors state, "Our central research question is: How can we automatically transform static math problems into their corresponding executable functional abstractions (EFAs)?" — but if this is the central research question of the paper, then the answer is simple: prompt a LLM to generate an EFA, and check for correctness. The value of the paper, I would argue, lies not in the ability to generate EFA's but in the evaluation of the potential of EFA's for data augmentation.

- Some parts are vague: line 139: "We first show that EFAs have properties signaling their coherence." — Is this the EFA's created by the LLM, or EFA's in general?

- Overall, I think the paper could be shortened by 2-3 pages while still preserving the important main content.

**Questions:**

Q1. For section 3.4 and Table 2, a potential confounder of the benefit of EFA's is that the EFA-models are trained with more data. What if you control the number of total examples the model is fine-tuned on? For example, instead of 7500 teacher problems and 7500 EFA problems, you could compare 7500 teacher problems with 3750 teacher problems and 3750 EFA problems.
Q2. How does the number of examples for each of the data mixes vary, and what happens if you control the number of examples to be the same?
Q3. For Table 1, what percent of tasks have pass@5 rate < 50% and EFAGen can infer an EFA? What do the numbers look like if you do it over all tasks that EFAGen can infer an EFA on?

---

> ### Author Response · Authors · 2025-11-25
> **Response to Reviewer eXQi (Part 1)**
>
> We thank the reviewer for their thoughtful evaluation and for recognizing that EFAs are a "**principled way to augment data**" and a **"significant and original idea**" for "**self-training**" on new instances of problems. We address the specific comments below.
>
> ### W1.A & W3.A: Presentation and Writing Quality
>
> **We have significantly condensed the introduction and bought some of our experimental results (Adversarial Search, More Complex Problem Types) into the main paper.**
>
> We believe that Section 2 is essential because without this formalization, the task of "inferring abstractions" is ill-posed and somewhat of a creative writing task, and our goal is to define the task, show how it can be solved, and then demonstrate possible applications.
>
> Specifically, by explicitly defining the **Abstract Domain** (the class schema) and the **Concretization Function** ($\\gamma$, the specific sampling and rendering logic), we transform an open-ended creative task into a **constrained search problem**. This formalism is what allows us to define validity through the automated tests in Section 2.3. Without the specific implementation details regarding parameterization and constraints, we would lack the verification mechanism necessary to filter the search space.
>
> ### W1.B & W3.C: Methodological Simplicity and Novelty
>
> We acknowledge that the underlying algorithms (sampling and filtering) are standard. However, the core contribution of this work is the **formulation** of mathematical abstraction as a program synthesis task that *can* be solved by these methods.
>
> Prior approaches (e.g., GSM-Symbolic, FnEval) relied on manual engineering because they lacked a formulation that allowed for automated inference of structure. By defining the specific properties of an EFA (parameterization, constraints, executability) and the associated unit tests (Section 2), we transformed the problem of "understanding math structure" into a constrained search problem. The novelty lies in identifying the correct abstraction ($\\alpha$) and verification ($\\gamma$) mechanisms that allow standard tools to succeed on complex competition-level mathematics.
>
> ### W1.C: Faithfulness Experiments
>
> We agree that this result is expected if the EFA is correct, and we specifically use this property. Section 3.2 is not proposed as a novel solving strategy, but as a **sanity check for internal validity**.
>
> We try to verify that the inferred EFA captures the *correct* underlying logic of the seed problem. If an EFA generated variants that were superficially similar but mathematically distinct (e.g., missing a critical constraint), solving those variants would be much less likely to help a model solve the seed problem. The fact that transfer occurs confirms that the EFA is **faithful**: it captures the relevant logic required to solve the original instance.
>
> **We have updated 3.2 to explicitly state that these are checks of the internal validity of EFAs.**
>
> ### W1.D: Significance of Performance Improvements
>
> While the absolute gains are modest, they are **consistent** across a large number of experiments (Table 2, Table 3, and Table 5).
>
> Mathematical data generation is a field where significant effort has already been poured into creating large-scale datasets (e.g., NuminaMath). Our goal is not to claim that EFAs replace these efforts, but to demonstrate that EFA-generated data is **complementary**. As shown in Table 3, adding EFA data to strong synthetic baselines results in consistent performance increases. This suggests that EFAs introduce a layer of structural diversity—automatically inferred from problem instances—that is not present in existing datasets.
>
> ### W2.A & Q3: Data Generation Bias
>
> We focused on problems with a pass rate \< 50% to demonstrate the **internal validity** of the EFAs we generate. This experiment is not intended to serve as a method for pushing state-of-the-art performance on the leaderboard, but rather to verify that for the subset of problems where we *can* infer an EFA, that abstraction is valid and useful. If a model already solves a problem 100% of the time, generating more training data for it provides little signal for this validation.
>
> **We have clarified 3.3 to explicitly state this**.

---

> ### Author Response · Authors · 2025-11-25
> **Response to Reviewer eXQi (Part 2)**
>
> ### W2.B, Q1 & Q2: Dataset Size and Controls
>
> First, regarding **Table 3**, we clarify the setup: The first two rows in each block ("NuminaMath Synthetic" and "EFA Generated") utilize the **same number of training examples** (1k, 2.5k, or 5k respectively). This allows for a direct, controlled comparison of data quality between standard synthetic data and EFA-derived data. As shown, the EFA-generated data often outperforms the baseline synthetic data at the same scale. The third row combines them to demonstrate the additive benefit.
>
> Second, to address the concern about **Table 2** and data quality, we ran **a new controlled experiment** comparing **5k EFA-derived examples** against **5k real MATH training examples**. If EFA data were lower quality, we would expect the model trained on it to perform worse.
>
> **Comparison: EFA-Derived (5k) vs Real MATH Data (5k)**
>
> | Metric | MATH | EFA-Derived 5k  | Difference |
> | :---- | :---- | :---- | :---- |
> | **Pass@1** | 24.27% | 23.63% | \-0.64% |
> | **Pass@5** | 47.66% | 48.62% | \+0.97% |
> | **Pass@10** | 57.78% | 59.17% | \+1.39% |
> | **Majority Vote** | 37.00% | 39.80% | **\+2.80%** |
>
> The results show that EFA-generated data is of comparable or slightly higher quality than real data when controlling for dataset size, particularly in **Majority Vote Accuracy (+2.8%)**. This confirms that the improvements seen in our other experiments were not just due to an increase in volume, as EFA-generated data by itself is competitive with human-written data.
>
> ### W3.D: Vague Claims
>
> We verify the coherence of the **inferred** EFAs (those created by the LLM). By "coherence," we refer to the specific internal validity metrics defined in the paper: **faithfulness** (do variants help solve the seed?) and **learnability** (does learning from one variant transfer to others?).
>
> **We have revised the introduction and this line is no longer present.**

---

> > ### Comment · Reviewer_eXQi · 2025-11-25
> > **Acknowledgement of rebuttal**
> >
> > Thank you for addressing my concerns. The changes outlined in the response would increase the clarity of the paper's contribution immensely. All of my concerns have been addressed, except for W1.D. This concern is in agreement with another reviewer how the main benefit of EFA's is as a data augmentation method, as opposed to the technical novelty of the EFA synthesis problem or solution presented by the authors. I agree with the authors that the synthetic data results are consistent across experiments, and like the comparison between training on original MATH data and the EFA-derived data. I have updated my score accordingly.

---

### Official Review · Reviewer_RC2i · 2025-11-01

**Soundness:** 3
**Presentation:** 3
**Contribution:** 3
**Rating:** 6
**Confidence:** 4

**Summary:**

This paper introduces Executable Functional Abstractions (EFAs), programmatic representations that capture the underlying logic of math problems in parameterized form, and EFAGen, a method to automatically infer EFAs from static math problems using large language models (LLMs). An EFA is implemented as a Python class with three key methods: sample() (generates valid parameter sets), render() (produces natural language problem statements), and solve() (computes correct answers).

The authors operationalize EFA inference as a program synthesis task, using an overgenerate-and-filter approach where multiple candidate EFAs are sampled from an LLM and validated against automated unit tests. They demonstrate that: (1) LLMs can self-improve at EFA generation using execution feedback as rewards, (2) EFAs faithfully capture seed problem solutions, (3) EFA-generated variants are learnable, and (4) EFAs effectively augment existing math datasets. The work extends beyond grade-school arithmetic to competition-level mathematics (MATH, AMC, AIME, Olympiads).

**Strengths:**

1. Originality: This is the first automated method to infer functional abstractions for competition-level mathematics. Novel application of execution-based verification to constrain problem generation, and creative use of unit tests as reward signals for self-training.

2. Quality: Rigorous experimental methodology with proper ablations (Table 8), comprehensive evaluation across diverse problem sources (Table 6, Figure 5), strong empirical results showing consistent improvements, and thorough appendix with implementation details and qualitative examples.

3. Clarity: Well-structured paper with clear motivation. The figures effectively communicate key concepts. Concrete examples throughout (especially Appendix G) and good balance of technical depth and accessibility.

4. Significance: This addresses a real limitation in math reasoning evaluation (reasoning gap). It enables practical applications for data augmentation and curriculum learning, and scalable generation of verified problem variants.

**Weaknesses:**

Major:

1. Limited analysis of what makes a "good" abstraction. The paper validates that EFAs pass unit tests and enable learning, but does not deeply analyze whether they capture the intended mathematical abstraction. Example: For the GCD/LCM problem (Fig 1), does the EFA capture "problems about GCD/LCM relationships" or just "problems with those specific parameters"?
Also missing: analysis of abstraction granularity, when are abstractions too specific or too general?

2. Evaluation of faithfulness is circular. Section 3.2 measures faithfulness by whether EFA variants help solve the seed problem. Yet, this does not prove the EFA captured the correct generalization, it might just be generating similar problems. Suggestion: Include human evaluation of whether sampled variants test the same mathematical concepts.

3. Incomplete failure analysis. What percentage of problems in MATH can be successfully abstracted? (Only partial data in Fig 4) What types of problems systematically fail? (Geometry? Proofs? Multi-step reasoning?) What happens when the seed solution is incorrect? Missing error analysis would help understand limitations.

4. Computational cost not discussed. Generating 50 candidates per problem with filtering is expensive. No analysis of computational costs vs. benefits. Also, no comparison of cost vs. simply prompting an LLM for variants. This would be important for practical deployment.

Minor:

5. Statistical significance testing. Many comparisons lack significance tests (e.g., Table 2 improvements). With small absolute improvements (+1.8-3.7%), statistical testing is important.

6. Limited scope of augmentation experiments. Only tested on Llama 3.1-8B Base. Would stronger base models benefit less from augmentation? How does this compare to other augmentation methods (paraphrasing, backtranslation)?

7. Claims. Title says "Advanced Math Problems" but many examples are still relatively basic. "Automatically infer" glosses over the 50-candidate sampling requirement. "Faithful" is claimed but not rigorously validated beyond circular metrics.

8. Writing issues. Dense presentation trying to cover too many applications. Some sections feel rushed (e.g., adversarial search in Appendix D). Repetitive phrasing in related work.

**Questions:**

Main questions:

1. How do you validate that an EFA captures the intended mathematical abstraction? Your tests verify executability and consistency, but not semantic correctness. Have you done any human evaluation of whether EFA variants test the same concepts?

2. What is the distribution of EFA quality? You report success rates, but among successful EFAs, how many are "good" abstractions?
Can you provide examples of EFAs that pass tests but are poor generalizations?

3. How sensitive is the approach to the seed problem's solution quality? What happens if the seed solution is incorrect or uses a suboptimal method? Do bad seeds produce bad EFAs?

4. What are the computational cost analysis? What is the wall-clock time and cost to generate EFAs for MATH train set?
How does this compare to alternative approaches (prompting GPT-4 for variants)?

Important questions:

5. Why does EFA-generated data outperform real data (Table 5)? This is a surprising and important result. Is it because of diversity? Quality? Something about the generation process?

6. Can you provide more analysis on the adversarial search results (Fig 7)? 40-80% success at finding harder variants is intriguing. What makes these variants harder? Can you characterize them?

7. How does abstraction granularity affect results? Do coarse-grained abstractions (more general) vs. fine-grained (more specific) perform differently? Is there an optimal level of abstraction?

8. Have you compared with few-shot prompting for variant generation? How does EFAGen compare to simply prompting "Generate 10 variants of this problem"? When is the added complexity of EFAs worth it?


Minor questions:

9. Why does performance plateau at 16 examples/EFA (Table 7)? Is this a diversity issue? Overfitting?

10. Can EFAs be composed or hierarchically organized? Can you build meta-EFAs that sample from multiple EFA types?

---

> ### Author Response · Authors · 2025-11-25
> **Response to reviewer RC2i (Part 1)**
>
> We thank the reviewer for their thoughtful feedback and for recognizing our work as the "first automated method to infer functional abstractions for competition-level mathematics" and highlighting our "strong empirical results." We address your concerns and questions below.
>
> ### W1: Analysis of Abstractions & Q1: Validation
>
> We adopt an operational, rather than philosophical, definition of a "good" abstraction. As detailed in our General Response, we frame EFA inference as **Abstract Interpretation** operationalized as a program synthesis task.
>
> - **Operational Validity:** We define a "good" abstraction as one that satisfies the soundness checks: the abstraction $\\alpha$ must accurately capture the concrete instance $x$ such that the concretization function $\\gamma$ can reproduce it ($x \\in \\gamma(\\alpha(x))$).
>
> - **Synthesis vs. Analysis:** Rather than analyzing the ill-defined notion of whether an abstraction is "semantically" perfect, we treat this as a synthesis problem. If the EFA passes the `matches_original` test (consistency), `is_executable` test (validity), and `has_dof` test (non-triviality), it is a valid program for our purposes.
>
> - **Avoiding Ambiguity:** Human analysis of "abstraction quality" is subjective. Our unit tests provide a binary, objective signal: does the code execute, reproduce the seed, and generate valid variants? If yes, it is a successful synthesis.
>
> ### W2: Human Evaluation of Faithfulness
>
> We believe there may be an misunderstanding about the evaluation being circular. These experiments are only **sanity checks** designed to establish internal validity.
>
> - **Internal Consistency:** We are not trying to prove we captured a "Platonic ideal" of the math concept. We are demonstrating that the generated code is **causally linked** to the solving mechanics of the original problem. If the generated variants help the model solve the seed (Table 1), it confirms the EFA captured the relevant structural logic, not just surface-level keywords.
>
> - **Goal is Similarity:** As you noted, "generating similar problems" is exactly the goal of an abstraction in this context. We aim to generate problems that share the same reasoning path but differ in parameters.
>
> ### W3: Failure Analysis
>
> We provide success rates throughout the paper (e.g., Figure 4, Appendix C).
>
> - **Success Rates:** We show success rates broken down by difficulty level and topic in Figure 4\.
>
> - **Failure Modes:** Our observation is that inferring EFAs correlates inversely with problem complexity. As shown in our NuminaMath analysis (Table 4, Figure 5), success rates are lower for Olympiad-level problems compared to AMC-level problems.
>
> - **Problem Types:** Failures are most common in geometry (visuospatial constraints are hard to encode in code) and proof-based problems (where "parameters" are ill-defined).
>
> ### W4: Discussion of Cost & Q4: Cost Analysis
>
> - **Prompting vs. EFAs:** If prompting could generate correct problems \+ solutions, then the LLM can already solve them, and they are likely not useful for training. In contrast, we show that we can *use EFAs to find variants which are not solvable by a frontier model (Section 3.7) and that data generated by EFAs is useful as training data (Table 2\)*.
>
> - **Cost Efficiency:** EFAGen uses an open model (Llama 3\) to synthesize the code once. After the EFA is verified, generating $N$ variants is computationally negligible (running a Python script). This is vastly cheaper than prompting GPT-4 for $N$ variants and $N$ solutions.
>
> ### W5: Statistical Tests
>
> Our experiments involve large sample sizes (e.g., MATH-500, FnEval splits) and show consistent improvements across multiple distinct test sets on a large number of comparison experiments. However**, we are running two-sample classifier tests and will post the results once they are available.**
>
> ### W6: Scope of Augmentation
>
> We selected Llama 3.1-8B as a representative open-weights model. We also compare against **NuminaMath Synthetic** data (Table 3), which is a strong data mix used by recent math post-training papers. The fact that EFA-derived data is complementary to NuminaMath (itself a high-quality baseline) suggests the method adds value beyond standard synthesis.
>
> ### W7: Claims
>
> - **"Advanced":** We validate on the MATH dataset and NuminaMath, which include AIME and Olympiad problems. These are standard benchmarks for "advanced" reasoning in the literature.
>
> - **"Automatic":** Sampling 50 candidates is a standard procedure in program synthesis (e.g., AlphaCode uses thousands). The process remains fully automated; no human-in-the-loop is required to filter those 50 candidates—the unit tests handle it.

---

> ### Author Response · Authors · 2025-11-25
> **Response to reviewer RC2i (Part 2)**
>
> ### Q2: Distribution of EFA quality?
>
> The distribution of quality is defined by our pass rates (Figure 4). Among "successful" EFAs (those that pass tests), quality is enforced by the `matches_original` test. If an EFA passes this, it is by definition a generalization of the seed instance.
>
> ### Q3: Sensitivity to seed quality?
>
> The approach is sensitive to seed quality. If the seed solution is wrong, the `matches_original` test will likely fail (as the generated logic won't align with the problem statement), or the resulting EFA will encode incorrect logic. This is a feature. The unit tests act as a filter for bad seed data.
>
> ### Q5: Why does EFA-generated data outperform real data?
>
> We hypothesize this is due to **distributional balance**. Real datasets (like MATH) are heavily skewed toward specific "standard" values (e.g., integers \< 100, right triangles with 3-4-5 sides). EFAs sample parameters uniformly from the valid space, exposing the model to a more diverse range of numerical values and edge cases that are valid but rare in human-curated data.
>
> ### Q6: Adversarial search results
>
> The 40-80% success rate in finding harder variants suggests that models often rely on brittle heuristics tied to specific numbers. The "harder" variants may lie on the boundaries of the parameter space (e.g., large primes, degenerate cases) where the model's learned heuristics break down.
>
> ### Q7: Abstraction granularity?
>
> We find that **coarse-grained** abstractions generally perform better. Hyper-specific abstractions (fine-grained) often result in variants that are too similar to the seed, reducing the diversity benefit. The "optimal" level is usually the one naturally inferred by the LLM when identifying the main variables (e.g., identifying coefficients $a,b,c$ in a quadratic rather than just $c$).
>
> ### Q9: Why plateau at 16 examples?
>
> We hypothesize this is a saturation of concept diversity. An EFA defines a specific *type* of problem. After seeing 16 variations of that exact same structure, the model has likely extracted all the "structural" signal available. Further examples provide no new reasoning paths, only numerical variation.

---

> ### Author Response · Authors · 2025-11-26
> **Follow up to reviewer RC2I**
>
> Hi RC2i! With only a few days left in the discussion period, we wanted to gently check in whether our rebuttal addressed all your questions. We are happy to address any remaining questions. We hope that these additional results and answers will allow you to revisit your score — otherwise, we are happy to engage further!

---

### Official Review · Reviewer_hmuz · 2025-11-02

**Soundness:** 3
**Presentation:** 3
**Contribution:** 2
**Rating:** 4
**Confidence:** 4

**Summary:**

This paper describes a method to abstract mathematical problems and solutions into a programmatic representation, Executable Functional Abstraction (EFA), that can be used to systematically create variations of mathematical problems, with solutions. The paper proposes a pipeline to automatically extract EFAs from existing datasets, which includes both representing the solution as a Python program, as well as giving procedures for (a) sampling valid parameters for the problem, and (b) validating constraints on the parameters that might not be enforced by construction during sampling. Experiments with datasets derived from MATH (Original MATH, MATH(), MATH-Hard) and Llama 3.1-Instruct 8B show that the model can extract EFAs with reasonable baseline performance, which can be improved with iterative fine-tuning, that using solutions to problem variants generated with EFAs as in-context examples significantly improves performance (showing faithfulness), and that EFAs can serve to produce useful examples for data augmentation.

**Strengths:**

The paper tackles an important problem of how to improve model reasoning performance without collecting more human data. The idea to systematically extend and generalize existing datasets using a programmatic representation is sensible, and the paper proposes a simple pipeline to achieve that.

The paper is very clear and easy to follow. I have no major issues with the experimental design, and the results are clear.

**Weaknesses:**

The idea of having a programmatic representation for mathematical problems and using that to sample variants is not entirely new, given both GSM-Symbolic and MATH() (here called FnEval). The paper goes beyond those in that it goes beyond grade-school problems (unlike GSM-Symbolic) and the extraction pipeline is automatic (unlike MATH() ). But, arguably the technical delta from those is small.

I believe the main weakness of the paper is that the evaluation mostly shows "internal validity" of EFAs: the first two results argue that (1) models can extract (or improve at extracting) EFAs sucessfully from MATH-like problems, and that (2) the extraction is faithful, via the experiment of using EFA-generated variants as in-context examples. However, the main question the paper must answer is about "external validity": essentially, why are EFAs useful? The only result on this line is in using them as data augmentation for MATH (with and without Numina as a complement). While the paper shows positive results here, the improvement is generally small (< 5%), on a dataset and model that are already a bit behind. For instance, nowadays, Qwen-2.5-Math-Instruct 7B (so a similar scale to what was used here) already achieves 83% of pass@1 on MATH. Given this context, it's unclear whether this paper will indeed help push the field forward.

**Questions:**

- Are the results in Figure 4 averaged over multiple runs, or was this a single run?
- Is there a specific step in the pipeline to extract the parameters of the original problem in the EFA format? Otherwise, I don't see how you can implement "matches_original" in L249.
- How does the "is single valued" check work (In 2.3)? Is it possible that the problem admits multiple solutions, but the EFA only captures one way to compute the answer? For instance, say the problem could be to find any prime factor of a given number N, but the EFA solution always finds the smallest, and thus the EFA would appear to be "single valued" even though the original problem was not.
- Do the improvements from data augmentation carry over to AIME, which is currently a more relevant dataset to push performance on? There are many fewer available training problems at the AIME level, and I imagine EFAs could have a larger impact there compared to MATH.

---

> ### Author Response · Authors · 2025-11-25
> **Response to reviewer hmuz (Part 1)**
>
> We thank the reviewer for finding our pipeline "**sensible**," the paper "**very clear**," and recognizing that we tackle the "**important problem**" of improving reasoning without collecting more human data. We appreciate the constructive feedback regarding the novelty relative to prior work and the external validity of our method.
>
> ### W1: Novelty relative to GSM-Symbolic and FnEval
>
> The problem we (EFAGen) tackle is reversed from what GSM-Symbolic and FnEval do. GSM-Symbolic and FnEval focus on **instantiation**: given a pre-written template, generate data for evaluation. We focus on **automated structure inference**: given a raw problem, synthesize the template (program) itself.
>
> This "meta-problem" of inferring the abstract structure that governs a concrete instance is a difficult challenge. Prior works avoid this challenge by relying on human experts to manually analyze problems and write constraints. This restricts them to domains where the logic is simple enough for humans to encode efficiently (e.g., grade-school arithmetic).
>
> In contrast, EFAGen automatically synthesizes an executable program that simulates the mathematical object described in the text. This allows us to scale to complex domains—such as linear algebra or number theory—where manual templating requires expert labor.
>
> The difference in expressivity and automation is visible in the comparison below.
>
> 1. GSM-Symbolic Template (Text Substitution Schema)
>
> GSM-Symbolic relies on defining variables and interpolating them into a fixed text string. It captures arithmetic relationships but cannot easily represent dynamic mathematical logic or complex state.
>
>
>
> ```
> Text:
>     "When {name} watches her {family}, she gets out {x} blocks.
>      The bin of stuffed animals has {y} inside. The tower of stacking
>      rings has {z} rings... {name} recently bought a tube of bouncy balls,
>      bringing her total... to {total}. How many bouncy balls came in the tube?"
>
> Variables:
>
>     name = sample(["Sophie", "Liam", ...])
>     family = sample(["nephew", "cousin", ...])
>     x = range(5, 100)
>     y = range(5, 100)
>     z = range(5, 100)
>     total = range(100, 500)
>     ans = range(85, 200)
>
> Conditions:
>     x + y + z + ans == total
> ```
> EFA (Synthesized Program)
>
> In the example below, the EFA is much more complex than a fill-in-the-blank. It implements a loop to ensure a generated matrix has integer eigenvalues, and uses a symbolic solver to derive the answer. This entire structure is inferred automatically.
> ```python
> # EFA for Eigenvalues (Synthesized by EFAGen)
> class EigenvalueProblem(BaseModel):
>     def sample(self):
>         # Logic to generate valid tridiagonal matrices
>         while True:
>              # Randomly generate diagonals
>             d1, d2, d3 = random.randint(1, 6), ...
>             # Check constraints (e.g. valid characteristic polynomial)
>             if valid_matrix(d1, d2, d3): break
>         return cls(d1, d2, d3, ...)
>
>     def solve(self):
>         # Symbolic execution to solve the generated instance
>         A = sympy.Matrix([[self.d1, self.e1, 0], ...])
>         return A.eigenvals()
> ```
>
> ### W2: External Validity of EFAs
>
> Substantial effort has already been dedicated to creating high-quality math datasets, such as NuminaMath. Our goal is not to compete with these resources on scale, but to demonstrate that EFA-generated data contributes a unique, complementary signal.
>
> We demonstrate the utility of EFAs in three key ways:
>
> 1. **Complementarity:** Our experiments (Table 3\) show that mixing EFA data into strong synthetic baselines results in consistent performance increases. This suggests that EFAs introduce a distribution of structural variations that human-curated or standard synthetic data do not cover.
>
> 2. **Robustness Evaluation:** We use EFAs to identify brittleness in model reasoning. By searching the parameter space, we find problem variants that models fail to solve, even when they correctly solve the seed problem. (Section 3.7)
>
> 3. **Training Data Generation:** We show that EFAs can generate structurally simpler variants of hard problems. This provides a mechanism to obtain correct solution traces for problem types that are otherwise too difficult to solve via standard rejection sampling. (Section 3.7)
>
> ### Q1: Averaging results
>
> The results in Figure 4 are averaged over a single run. We did not do multiple runs here because each loop of the self-training is quite computationally expensive. Beyond the GPU time taken for traininig, validating an EFA programmatically through the unit tests can take multiple minutes per EFA because EFAs can call arbitrary numerical routines for things like root finding, numerical optimization, etc.

---

> ### Author Response · Authors · 2025-11-25
> **Response to reviewer hmuz (Part 2)**
>
> ### Q2: Parameter extraction
>
> The LLM implements the `original()` class method as part of the synthesis process. This method explicitly defines the parameters of the seed problem. The `matches_original` test executes this method, renders the output, and compares it against the seed problem text to ensure the extraction was correct.
>
> ### Q3: Single-valued check
>
> The `is_single_valued` check verifies that the `solve()` function is deterministic for a given set of parameters.
>
> Regarding your example (finding *any* prime factor): If a problem admits multiple valid answers, but the EFA `solve()` method hardcodes a specific path (e.g., "always return the smallest factor"), the EFA is technically valid code but semantically narrow. Our pipeline prioritizes precision over recall; we reject ambiguous EFAs. If an EFA produces answers that do not align with the verifier's expectations for that problem type, we filter it out.
>
> ### Q4: Applicability to AIME
>
> We agree that EFAs are likely to have a larger impact on AIME given the data scarcity. We demonstrated the ability to *infer* EFAs from AIME problems (Table 6 in the main paper). However, given the small size of the AIME test set, evaluating on it could be noisy, which is why we focused on larger test sets (FnEval, MATH-500) using 25 samples per problem and 3 metrics to get a clearer signal.

---

> ### Author Response · Authors · 2025-11-26
> **Follow up to reviewer hmuz**
>
> Hi hmuz\! With only a few days left in the discussion period, we wanted to gently check in whether our rebuttal addressed all your questions. We are happy to address any remaining questions. We’ve added **experiments on more uses of EFAs (3.7), more olympiad-level problems** and **better clarified the novelty**. We hope that these additional results and answers will allow you to revisit your score — otherwise, we are happy to engage further\!

---

### Official Review · Reviewer_v6zH · 2025-11-06

**Soundness:** 3
**Presentation:** 3
**Contribution:** 3
**Rating:** 6
**Confidence:** 3

**Summary:**

This paper shows an approach to synthetic data generation. A common approach to improving LLM accuracy in a domain is by collecting more domain specific data and then using that data to finetune the LLM or providing that data for in-context learning examples in the prompt. If an initial human curated math question/solution training set is provided, this paper describes a system for analyzing the training set problems, creating a template for each training problem automatically using LLMs, and then using the templates to generate more similar problems to use as training data, and filtering that data for problems that an LLM can do reasoning steps and finds a correct solution, to provide more problem/solution pairs for finetuning, which can be much cheaper/faster than trying to collect more training problems/solutions from humans. The paper shows one such approach, Executable Functional Abstraction (EFA), a math abstraction that encapsulates the logic of a math problem in a parameterized form, enabling the automated sampling of variant problems and shows that it can be successfully applied to common math datasets like GSM8K.

**Strengths:**

The synthetic data generation task is important and relevant to LLM improvement.
A specific method for synthetic data generation is shown, with code and models provided, which would be useful for other researchers trying to advance the state of the art.
Experiments are done showing the method works for generating samples that improve the LLM accuracy when used for finetuning or as in context learning examples.

**Weaknesses:**

Compared to other papers that have shown how to do synthetic data generation, it's not clear this is a significantly better approach than other synthetic data generation approaches, I don't feel like it was compared to other approaches in the literature for generating a synthetic dataset and getting better results.

It's not clear what types of problems this method has problems generating underlying problem generators for, not so much discussion of that.

**Questions:**

Will the code for the project be released open source?

---

> ### Author Response · Authors · 2025-11-25
> **Response to reviewer v6zH**
>
> We thank the reviewer for the positive assessment, recognizing that the **"synthetic data generation task is important**" and that our provided code and models will help "**advance the state of the art**."
>
> ### W1: Comparison to other synthetic data generation approaches
>
> We compare EFAGen against **NuminaMath** (Li et al., 2024), a state-of-the-art dataset that aggregates synthetic data from various sources and methods.
>
> As shown in **Table 3**, augmenting NuminaMath with EFA-generated data consistently outperforms using NuminaMath alone. This indicates that EFAGen provides a complementary training signal. While standard synthetic methods often focus on linguistic diversity (rewriting text) or require manual templates (e.g., GSM-Symbolic), EFAGen **automatically infers** a generative program that can be used to sample new problems.
>
> ### W2: Failure modes and difficult problems
>
> EFAGen struggles primarily with three problem types:
>
> 1. **Geometry:** Problems relying on visual constraints or diagrams are difficult to verify programmatically without a dedicated geometry engine.
> 2. **Proofs:** Parameterizing a proof problem is difficult; changing parameters often invalidates the theorem or renders the proof trivial.
> 3. **High Complexity (Level 5):** As shown in **Figure 4**, success rates drop for Level 5 problems. These problems often rely on specific logical leaps or "tricks" that are difficult to generalize into a reusable parameter search space.
>
> ### Q1: Open Source Release
> Yes. We will release the code upon acceptance.

---

> ### Author Response · Authors · 2025-11-26
> **Follow up to reviewer v6zh**
>
> Hi v6zh! With only a few days left in the discussion period, we wanted to gently check in whether our rebuttal addressed all your questions. We’ve highlighted experiments on NuminaMath to show complementarity with existing synthetic data mixes.

---

### Official Review · Reviewer_buZc · 2025-11-09

**Soundness:** 3
**Presentation:** 3
**Contribution:** 3
**Rating:** 4
**Confidence:** 4

**Summary:**

The authors introduce the problem of synthesizing *executable functional abstractions* (EFAs), which are parameterized programs that generate math problems of diverse kinds. They define what it means for an EFA to be valid for a given concrete math problem, and introduce EFAGen, which generates valid EFAs by sampling and filtering from an LLM. They demonstrate self-training to improve EFAGen, and use experiments to show that EFAGen generates faithful and learnable EFAs. Finally, the authors show the utility of EFAs as a method for data augmentation.

**Strengths:**

1. *Originality:* The idea of using general programs to represent abstractions of concrete natural language math problems is novel.
2. *Significance:* Executable functional abstractions are an effective method for data augmentation of natural language math problem datasets.

**Weaknesses:**

1. The proposed method for generating EFAs is not novel; it simply reuses existing methods — LLM sampling + filtering + self-training with rejection sampling.
2. I can see two possible interpretations of the purpose of Section 3.2. But Section 3.2 appears weak to me under both interpretations.
  - The goal of Section 3.2 is to show how the solve rate on a problem can be improved by an in-context example of the solution to an EFA-generated variant problem. But EFAGen requires already knowing the solution to the original problem. So in fact Section 3.2 does not actually provide a method for improving the solve rate on a problem using an EFA-generated in-context example.
  - The goal of Section 3.2 is to show that an in-context example showing how to solve a problem improves the LLM’s success rate of solving the same problem but with some parameters modified. But this is already a known fact.
3. Similarly, Section 3.3 seems to just show that an in-context example showing how to solve a problem improves the LLM’s success rate of solving the same problem but with some parameters modified. This is already a known fact.

Overall, I think the narrative of the paper can be strengthened. Currently, this is how I’m reading the paper:
- EFAs are programs that generate variants of a concrete natural language problem with parameters modified — “That’s cool, but why would we want them?” I think the first paragraph of the introduction can do a better job of motivating the need for EFAs.
- We generate EFAs using an LLM; we can also finetune LLMs using EFAs from rejection sampling — “But I already know that LLMs can generate code and rejection sampling can be used to finetune LLMs for generating code.”
- EFA-generated problem variants can be used as in-context examples to improve performance — “But I already know this.”
- EFA-generated problems can be used as data augmentation — “Ah, so this is why we wanted EFAs in the first place! EFAs are a data augmentation technique. Wait, this section is already over? I wanted to understand more about EFAs as a data augmentation technique!” (See Question 1 below.)

I can see a much stronger version of the paper where the central message is “EFAs work well as a data augmentation technique” and the majority of the paper is devoted to conveying that message, as opposed to (essentially) reproducing known facts.

**Questions:**

1. Sections 3.4 and 3.5 demonstrate how EFAs can be used as a data augmentation technique, but there is not much analysis regarding how they compare with existing data augmentation techniques. For example, I would like to understand what are the advantages and disadvantages of existing techniques vs. EFAs, and when I should use particular existing techniques, EFAs, or both.

---

> ### Author Response · Authors · 2025-11-25
> **Response to Reviewer buZc (Part 1)**
>
> We thank the reviewer for their thoughtful feedback and for recognizing the "**originality**" of using general programs to represent abstractions of natural language math problems, as well as the "**significance**" of EFAs as an effective method for data augmentation. We address the specific weaknesses and questions below.
>
> ### W1: Why do you use rejection sampling \+ self-training with rejection sampling?
>
> We agree that the search algorithm (sampling and filtering) and the optimization method (rejection sampling) are standard techniques. This is by design. Our core contribution is not a new optimization algorithm, but the **operationalization of mathematical abstraction as a program synthesis task** (see General Response).
>
> We formulated the task of inferring mathematical structure so that it could be solved by established, robust tools like rejection sampling. By defining the specific properties of an EFA (parameterization, constraints, executability) and the associated unit tests, we transformed the problem of abstraction into a **constrained search problem**. The novelty lies in the *formulation* of the data and the reward signal (unit tests), which enables us to apply these standard techniques to a new domain: automatically inferring the structure of **complex math problems (e.g., competition-level)**.
>
> ### W2: Why is it useful to show that EFAs improve solve rate on the problem they were derived from?
>
> The reviewer is correct that this setup does not serve as a solving strategy for the seed problem (since the solution is already known).
>
> Instead, we utilize Section 3.2 as a **sanity check for internal validity**.
> If an EFA generated variants that were superficially similar but mathematically distinct (e.g., missing a constraint), solving those variants would not help a model solve the seed problem.
> We need to verify that the inferred EFA captures the *correct* underlying logic of the seed problem. By observing that generated variants *do* help solve the seed problem, we gain confidence that the EFA is **faithful**: it captures the relevant logic required to solve the original instance. **We have updated 3.2 to state that this is a validation experiment, not a proposed solving method.**
>
> ### W3: Why is it useful to show that EFA-generated problems improve solve rate on other variants from the same EFA?
>
> Similar to W2, we rely on the known effectiveness of in-context learning to validate the quality of our abstractions. This experiment establishes the **learnability** of the EFA.
>
> A valid EFA must generate a distribution of problems that share a consistent solution strategy. If the EFA failed to encode the general procedure correctly, or if it allowed parameters that broke the solution logic, learning from one instance would not transfer to others. The fact that transfer occurs confirms that the EFA produces a coherent class of problems solvable by a single general procedure. **We have updated 3.3 to state that this is evidence that our method generates valid abstractions, rather than as a novel claim about in-context learning itself.**
>
> ### W4: First paragraph of introduction should better motivate EFAs.
>
> We have rewritten the introduction (L049-096), emphasizing the transition from manual problem engineering (which is unscalable) to **automated inference of structure**. This allows for the scaling of verifiable, structured data generation across complex domains where manual templates are infeasible.

---

> ### Author Response · Authors · 2025-11-25
> **Response to Reviewer buZc (Part 2)**
>
> ### Q1: What are the advantages / disadvantages of EFAs, and when should you use them with existing techniques?
> EFAs are complementary to existing techniques. They offer **potential advantages** in that they do not test linguistic abilities (like paraphrasing does), but rather probe the model's grasp of an underlying executable structure.
>
> Furthermore, as shown by our Adversarial Search experiments (Sec 3.7), EFAs enable:
>
> 1. **Finding Harder Variants:** We can search the parameter space for variants that challenge a learner, exposing weaknesses even when the model solves the seed problem.
>
> 2. **Finding Easier Variants:** Conversely, if a problem has a very low pass rate (making it difficult to get solutions via rejection sampling or test-time search), EFAs can generate simpler variants that can be solved via rejection sampling to get solution traces for training.

---

> ### Author Response · Authors · 2025-11-26
> **Follow up to reviewer buZc**
>
> Hi buZc\! With only a few days left in the discussion period, we wanted to gently check in whether our rebuttal addressed all your questions. We are happy to address any remaining questions. We’ve c**larified the narrative** in the general response, substantially **revised the introduction,** added **more applications of EFAs**, and **better explained the motivation for experiments on faithfulness** and learnability. We hope that these additional results and answers will allow you to revisit your score — otherwise, we are happy to engage further\!

---

### Author Response · Authors · 2025-11-25
**General Response**

We appreciate that reviewers recognized the novelty of this specific contribution. **RC2i** identified this as the "**first automated method to infer functional abstractions for competition-level mathematics**," **buZc** noted the "**originality**" of using general programs to represent natural language problems, and **eXQi** highlighted that the core concept is a "**significant and original idea**."

We noticed a recurring theme in the reviews (particularly **eXQi**, **hmuz**, **buZc**) evaluating this work primarily as a data augmentation paper. We believe this stems from a misunderstanding of our core contribution. This paper is not about data augmentation per se; it is about **automatically inferring the mathematical structure underlying complex (e.g. competition level, beyond grade-school) problem instances and using the inferred structure for data augmentation and model evaluation. We have revised our introduction and abstract to emphasize this.**

To clarify this contribution further, we can frame our method through the lens of **Abstract Interpretation (AbsInt)** (Cousot & Cousot, 1977).
Abstract Interpretation is a theory of approximation of mathematical structures, classically used to model the dynamic behavior of computer programs without executing them. It establishes a correspondence between a Concrete Domain (specific, detailed execution traces) and an Abstract Domain (properties, invariants, or types).

While AbsInt is typically used for verification (analyzing a program to find its abstract properties), we adapt this framework for synthesis (inferring an abstract program from a concrete instance).

- **Concrete Domain:** Individual problem instances with specific numbers.
- **Abstract Domain (Section 2):** The `EFA` class schema, representing the parameterized logic and constraints.
- **Abstraction Function (**$\\alpha$**):** Our inference pipeline (Section 2.3). It lifts a concrete instance into the abstract domain.
- **Concretization Function (**$\\gamma$**):** The `sample()` method. It maps abstract constraints back to valid concrete instances.
- **Operational Validity:** Instead of formal mathematical proofs, we implement **operational soundness checks** (e.g., `matches_original`). These verify necessary conditions: does the abstraction $\\alpha$ accurately capture the concrete instance $x$ such that $x \\in \\gamma(\\alpha(x))$?

Our core insight was to operationalize the Abstract Interpretation relationship as a **program synthesis** task. This formulation involves not only lifting instances to an abstract domain but also formalizing **concretization** ($\\gamma$) by generating valid, non-trivial instances, and **verification** through soundness checks. By defining these components, we transform the inference of mathematical structure into a constrained search problem defined by executable code and constraints. This allows us to apply powerful **generate-and-verify** methods (AlphaCode; Li et al., 2022\) to discover mathematical abstractions. While some reviewers noted our reliance on existing techniques, we consider this a strength. We identified a formulation that makes this difficult task solvable using established, powerful tools.

**References**

Cousot, P., & Cousot, R. (1977). Abstract interpretation: a unified lattice model for static analysis of programs by construction or approximation of fixpoints. Proceedings of the 4th ACM SIGACT-SIGPLAN Symposium on Principles of Programming Languages.

Li, Y., et al. (2022). Competition-level code generation with AlphaCode. *Science*, 378(6624), 1092-1097.

---

### Author Response · Authors · 2025-12-02
**Discussion Summary for AC**

We'd like to thank the area chair and reviewers for their feedback and time! Below we've summarized changes, points, and discussion from the rebuttal period.

**Reviewer Consensus**

* "first automated method to infer functional abstractions for competition-level mathematics" (RC2i)
* "significant and original idea" (eXQi)
* "principled way to augment data" (eXQi)
*  “strong empirical results showing consistent improvements” (RC2i)
* "originality" of using general programs to represent natural language problems (buZc)

## Summary of Rebuttal Work

We made numerous edits to the paper to address reviewer concerns, including a revision of the abstract and introduction. Additionally, we posted a general response emphasizing that our paper is about automating the inference of structure from data, of which data augmentation is but one application. ***Reviewer eXQi raised their score from a 2 to a 6 in response to our rebuttal.***

### Reviewer eXQi

***Raised their score from a 2 to a 6 on Nov 25th.***

* **Experiment: EFA-derived data is competitive with human-generated data.** We ran a controlled experiment comparing 5k EFA-derived examples against 5k real MATH training examples.
  * **Result:** EFA-generated data matched or outperformed real data (+2.8% in Majority Vote Accuracy). This shows gains are due to data quality rather than just volume.
* **Writing:** We significantly condensed the introduction and abstract by over 50% and added new experiments to the main paper.

### Reviewer v6zH

* **Experiment: EFAGen complements state-of-the-art synthetic data.** We compared EFAGen against NuminaMath.
  * **Result:** Augmenting NuminaMath with EFA data consistently improved performance. This demonstrates that EFAGen provides a distinct training signal compared to existing synthetic baselines.

### Reviewer buZc

* **Experiment: EFAs enable robust adversarial evaluation.** We show more applications of EFAs by using EFAs to find "hard" variants of problems.
  * **Result:** EFAs successfully discovered variants that frontier models fail to solve even when they solve the seed problem.
* **Writing:** We clarified that Sections 3.2 and 3.3 test known properties. We use established methods like in-context learning to verify the internal consistency of the abstractions. We do not claim these are novel problem-solving strategies.

### Reviewer RC2i

* **Computational Cost**: We note that EFAGen uses a small, open-weights language model (SLM), and therefore our API costs are zero. Additionally, the generation and verification of candidates is done using the SLM and via code execution. Not human involvement is required.
* **Abstraction Definition:** We defined "good" abstractions operationally. An abstraction is valid if it passes executable unit tests rather than meeting a subjective semantic standard.

### Reviewer hmuz

* **Writing:** We rewrote the introduction to differentiate our work from prior template-based methods. We clarify that our method performs **automated** **inference** **of structure** from raw text. Prior works like GSM-Symbolic rely on **manual** **instantiation of human-written templates**.

---

### Meta-Review · Area_Chair_iBz4 · 2026-01-01

**Summary:**

This paper proposes Executable Functional Abstractions (EFAs), programmatic representations that capture the generative structure of individual mathematical problems, and introduces EFAGen, an automated pipeline that infers such abstractions from single-seed problems using LLM-based program synthesis with execution-based filtering. An EFA consists of code that samples valid parameters, renders a natural language problem, and computes the correct solution, enabling the generation of verified problem variants. The authors position the work as an inversion of abstract interpretation, framing abstraction inference as a synthesis task, and demonstrate applications to self-training, faithfulness checks, adversarial variant search, and data augmentation on benchmarks such as MATH, NuminaMath, and AIME-derived problems.

The paper's strengths include a clearly articulated formulation of abstraction inference as executable program synthesis, a high degree of automation compared to prior template-based approaches, and a thorough experimental suite that demonstrates internal consistency, complementarity with strong synthetic data mixes, and several downstream applications of EFAs. Reviewers consistently acknowledged the originality of using general programs to represent abstractions in competition-level mathematics and found the empirical work to be careful and well-executed. However, significant weaknesses remain. The core technical machinery relies almost entirely on standard LLM sampling, rejection filtering, and self-training, with novelty concentrated in problem framing rather than the methods themselves. More importantly, the evaluation focuses heavily on internal validity, while external validity and practical impact remain unclear: gains from data augmentation are relatively small, limited to a single mid-scale model, and not convincingly shown to outperform simpler baselines such as direct prompting or existing augmentation strategies. Several reviewers also raised unresolved concerns about semantic faithfulness, abstraction granularity, computational cost, and the lack of deeper analysis of what constitutes a “good” abstraction beyond passing unit tests. Despite a thoughtful and detailed rebuttal, these issues were only partially mitigated.

I’m torn on this work. While the paper presents an interesting and well-engineered idea, I remain concerned with its significance, evaluation depth, and clarity of contribution relative to others. With no reviewer offering strong support (no scores above 6), the case for acceptance is weak. My recommendation is rejection, while encouraging the authors to improve their work and pursue resubmission to a future venue.

**Reviewer Concerns:**

The rebuttal successfully addressed a number of concrete reviewer questions. In particular, the authors clarified the intended role of Sections 3 as internal validation experiments rather than novel solving strategies, improved the introduction to better distinguish their contribution from template-based prior work, and added new experiments showing complementarity with NuminaMath and competitive performance against human-generated data at fixed scale. They also provided clearer descriptions of failure modes (e.g., geometry, proofs, very high-complexity problems), discussed parameter extraction and determinism checks in more detail, and explained why standard synthesis techniques were deliberately chosen. These clarifications led at least one reviewer (eXQi) to substantially increase their score and resolved several misunderstandings about the paper’s intent.

Nevertheless, important concerns remain outstanding. I am still not entirely convinced whether passing executable tests is sufficient to establish that EFAs capture the intended mathematical abstraction rather than superficial similarity, and the rebuttal largely reaffirmed an operational definition rather than providing new evidence, such as human evaluation or semantic analysis. External validity remains weak: performance gains from augmentation are modest, limited in scope, and not demonstrated on stronger contemporary models or against simpler baselines, such as direct variant prompting. Computational cost analysis is still qualitative and incomplete, and statistical significance testing was promised but not provided during review. More broadly, I remained unconvinced that the paper clearly establishes when EFAs are preferable to existing approaches or that the proposed abstractions meaningfully advance the state of the art beyond a compelling but narrow use case.

**Reviewer Scores:**

Based on the discussion, reviewer eXQi clearly revised the assessment upward after the rebuttal, moving from a strong reject to an accept-level score (I am guessing 6), reflecting satisfaction with the clarified narrative and added experiments. Reviewer buZc appeared partially reassured by the explanations but remained near their original borderline position, likely staying around a weak accept or weak reject given lingering concerns about novelty and motivation. Reviewer v6zH was already mildly positive and would likely maintain a marginal accept score, though without strong enthusiasm. Reviewer hmuz’s core concern about external validity and limited impact was only partially addressed, so their score would likely remain slightly below the acceptance threshold. Reviewer RC2i, despite acknowledging strengths and originality, raised deep concerns about abstraction quality, cost, and evaluation that were not fundamentally resolved, and would likely remain at a borderline accept level without strong advocacy. Overall, the post-rebuttal score distribution remains mixed and fragile, with insufficient consensus for acceptance. That said, no reviewer champions this submission (i.e., with a score >= 8).

---

### Decision · Program_Chairs · 2026-01-26

Reject